# Towards Reliable Model Selection for Unsupervised Domain Adaptation: An Empirical Study and A Certified Baseline

**Dapeng Hu**[1][*]    **Mi Luo**[3]    **Jian Liang**[4,5][†]    **Chuan-Sheng Foo**[2,1]

[1]Centre for Frontier AI Research, A*STAR, Singapore
[2]Institute for Infocomm Research, A*STAR, Singapore
[3]National University of Singapore
[4]NLPR & MAIS, Institute of Automation, Chinese Academy of Sciences
[5]School of Artificial Intelligence, University of Chinese Academy of Sciences

## Abstract

Selecting appropriate hyperparameters is crucial for unlocking the full potential of advanced unsupervised domain adaptation (UDA) methods in unlabeled target domains. Although this challenge remains under-explored, it has recently garnered increasing attention with the proposals of various model selection methods. Reliable model selection should maintain performance across diverse UDA methods and scenarios, especially avoiding highly risky worst-case selections—selecting the model or hyperparameter with the worst performance in the pool. *Are existing model selection methods reliable and versatile enough for different UDA tasks?* In this paper, we provide a comprehensive empirical study involving 8 existing model selection approaches to answer this question. Our evaluation spans 12 UDA methods across 5 diverse UDA benchmarks and 5 popular UDA scenarios. Surprisingly, we find that none of these approaches can effectively avoid the worst-case selection. In contrast, a simple but overlooked ensemble-based selection approach, which we call EnsV, is both theoretically and empirically certified to avoid the worst-case selection, ensuring high reliability. Additionally, EnsV is versatile for various practical but challenging UDA scenarios, including validation of open-partial-set UDA and source-free UDA. Finally, we call for more attention to the reliability of model selection in UDA: avoiding the worst-case is as significant as achieving peak selection performance and should not be overlooked when developing new model selection methods. Code is available at https://github.com/LHXXHB/EnsV.

## 1   Introduction

Deep learning has achieved incredible advancements in various tasks through supervised learning with large labeled datasets [1]. However, obtaining labels can be expensive, and deep models often struggle to generalize to unlabeled data from unseen distributions [2]. Domain adaptation [3] tackles this challenge by transferring knowledge from a labeled source domain to a target domain with limited labels but a similar task. Unsupervised domain adaptation [4] (UDA), particularly, has garnered significant attention due to its practical assumption that the target domain is entirely unlabeled, witnessing the development of many effective methods [5–8] and practical settings [9–12].

However, successful applications of UDA methods across diverse tasks rely heavily on selecting appropriate hyperparameters. Sub-optimal hyperparameters can cause state-of-the-art UDA methods to

---

[*]This work was completed while Dapeng (`lhxxhb15@gmail.com`) was a scientist at A*STAR.
[†]Corresponding author: Jian Liang (`liangjian92@gmail.com`)

38th Conference on Neural Information Processing Systems (NeurIPS 2024) Track on Datasets and Benchmarks.

Table 1: Statistics for worst-case selections by various model selection methods are provided across 110 closed-set UDA tasks (potentially an additional 21 tasks on DomainNet [13]), 24 partial-set UDA tasks, and 17 source-free UDA tasks (only for applicable methods). These statistics represent the count of worst-case selections divided by the total count of tasks, with **bold** font indicating the best worst-case avoidance. 'n.a.' indicates that certain methods are not applicable without source data.

| Method | Closed-set UDA | Partial-set UDA | Source-free UDA |
|---|---|---|---|
| SourceRisk [9] | 16 / 110 | 2 / 24 | n.a. |
| IWCV [14] | 15 / 110 | 3 / 24 | n.a. |
| DEV [15] | 9 / 110 | 1 / 24 | n.a. |
| RV [16] | 2 / 110 | 1 / 24 | n.a. |
| Entropy [17] | 15 / 131 | 7 / 24 | 16 / 17 |
| InfoMax [18] | 9 / 131 | 12 / 24 | 16 / 17 |
| SND [19] | 33 / 131 | 3 / 24 | 11 / 17 |
| Corr-C [20] | 80 / 131 | 4 / 24 | 3 / 17 |
| EnsV (Ours) | **0 / 131** | **0 / 24** | **0 / 17** |

underperform compared to the source-trained model without target-domain adaptation [19, 18]. This phenomenon emphasizes the significance of model selection, also called hyperparameter selection or validation, in UDA. Taking the typical one-hyperparameter validation task of a given UDA method as an example, we need to determine the optimal value of a hyperparameter $\eta$ among a set of $m$ different candidate values $\{\eta_i\}_{i=1}^m$. By applying these different $\eta_i$ with the same UDA method, we can obtain a set of $m$ different models with the parameter weights $\{\theta_i\}_{i=1}^m$. The goal is to identify the candidate model that exhibits the best performance on the unlabeled target domain and subsequently adopt the associated hyperparameter value for $\eta$. This model selection problem remains challenging and under-explored in UDA due to cross-domain distribution shifts and the absence of labeled target data.

Existing approaches can be categorized into two types. The first type involves leveraging labeled source data for target-domain model selection [9, 14–16]. The second type designs unsupervised metrics based on priors of the learned target-domain structure and utilizes the metrics for model selection [17, 19, 18, 20]. It is natural to ask: Are these approaches reliable in model selection tasks, i.e., can they maintain good performance for various practical UDA tasks?

To answer this question, we conduct an extensive empirical study to assess the performance of all selection methods across various practical UDA settings, including closed-set UDA [21], partial-set UDA [10], open-partial-set UDA [11], and source-free UDA [12, 22]. Notably, the model selection problem of open-partial-set UDA has not been investigated before. Surprisingly, we find that despite their specific designs, all these methods encounter challenges in avoiding the selection of poor or even the worst models across various UDA methods and settings. This renders the adaptation ineffective or even harmful, thereby constraining their adoption by researchers and practitioners in the community [18]. For instance, Table 1 compares the worst-case selection statistics of all these model selection methods across various practical UDA settings. These settings include standard closed-set UDA and partial-set UDA, which have been extensively studied in prior works [15, 19], and source-free UDA, where the model selection problem has not been widely investigated. The comparison reveals that all the methods occasionally or even frequently suffer from worst-case model selection situations, indicating high unreliability.

In contrast, we note that a simple ensemble-based validation baseline, dubbed EnsV, can effectively avoid the worst-case selection. Through a straightforward theoretical analysis of the ensemble, we observe that it is guaranteed to surpass the worst candidate model's performance. Our introduced EnsV takes a further simple step, utilizing the ensemble as a role model for directly assessing candidate models during the model selection process. This strategy ensures the secure avoidance of selecting the worst candidate model, thereby enhancing the reliability of model selection. Moreover, EnsV only uses target-domain predictions inferred by all candidate models. This eliminates the need for specific domain shift assumptions and access to source data, while also requiring no additional effort, such as time and memory, as all models are provided within the given problem context. This simplicity and versatility make EnsV suitable for various practical UDA scenarios, including the unexplored challenges of validation for UDA with unknown open classes [19]. Despite EnsV not being certified for peak-performance selection, we hope that, as the first to focus on the practical

Table 2: Comparisons of unsupervised model selection approaches used for UDA.

| Method | covariate shift | label shift | w/o source data | w/o extra hyperparameter | w/o extra training | worst-case avoidance |
|---|---|---|---|---|---|---|
| SourceRisk [9] | ✗ | ✗ | ✗ | ✗ | ✓ | ✗ |
| IWCV [14] | ✓ | ✗ | ✗ | ✗ | ✗ | ✗ |
| DEV [15] | ✓ | ✗ | ✗ | ✗ | ✗ | ✗ |
| RV [16] | ✓ | ✗ | ✗ | ✗ | ✗ | ✗ |
| Entropy [17] | ✓ | ✗ | ✓ | ✓ | ✓ | ✗ |
| InfoMax [18] | ✓ | ✗ | ✓ | ✓ | ✓ | ✗ |
| SND [19] | ✓ | ✓ | ✓ | ✗ | ✓ | ✗ |
| Corr-C [20] | ✓ | ✗ | ✓ | ✓ | ✓ | ✗ |
| EnsV (Ours) | ✓ | ✓ | ✓ | ✓ | ✓ | ✓ |

aspect of worst-case avoidance in model selection, our empirical study and simple baseline can inspire future efforts in developing more reliable model selection methods.

## 2 Related Work

**Unsupervised domain adaptation** (UDA) is initially studied in a closed-set setting (CDA) where only covariate shift [14] is considered as the domain shift, and the two domains share the same label set. Recent research has explored many real-world UDA scenarios by incorporating label shift, where the two domains have distinct label sets. This includes partial-set UDA (PDA) [10], where several source classes are missing in the target domain, open-set UDA (ODA) [23], where the target domain contains samples from unknown classes, and open-partial-set UDA (OPDA) [11], where there are only some overlaps in the label sets across domains. More recently, source-free UDA settings (SFUDA) [24, 12] have been explored, where only the source model instead of source data is available for target adaptation, potentially addressing privacy concerns in the source domain. Subsequently, in the context of black-box domain adaptation [22], the privacy of the source domain is fully safeguarded. Specifically, the research community has made significant efforts to develop effective UDA methods in image classification [9, 6] and semantic segmentation [25, 26], which can be seen through two distinct research directions. The first direction focuses on aligning the distributions across domains by minimizing specific discrepancy measures [27, 28, 21, 29, 30] or using adversarial learning to maximize domain confusion [9]. Especially, adversarial learning has become a popular approach and has been explored at different levels for domain alignment, including image-level [31], manifold-level [9, 32, 6], and prediction-level [5, 25, 26, 33]. The second direction focuses on target-oriented learning, aiming to learn a good structure for the target domain. This includes self-training approaches [34, 12, 35] and target-specific regularizations [7, 8, 36]. To thoroughly assess the efficacy of model selection baselines, we opt for a diverse set of UDA methods across various UDA scenarios in our model selection experiments and then utilize these baselines to choose the appropriate hyperparameters for different UDA methods.

**Model selection** for out-of-distribution (OOD) testing data is crucial for practical model deployment, but it remains challenging. Although the problem has attracted increasing attention in both domain generalization (DG) [37, 38] and UDA [18, 19], it remains relatively under-explored. In DG, since target data is not available for model selection, existing methods usually estimate the general OOD performance with multiple source domains. Differently, in UDA, thanks to the transductive setting, target data can be used for model selection in various ways. Efforts to address UDA model selection can be broadly categorized into two lines. Early approaches focused on estimating the target domain risk through labeled source data. SourceRisk [9] utilized a hold-out labeled source validation set to guide model selection based on source risk. To mitigate the impact of domain shift on source estimation, [14] introduced Importance-Weighted Cross-Validation (IWCV), which re-weights source risk using a source-target density ratio estimated in the input space. Building upon this, [15] improved IWCV by introducing Deep Embedded Validation (DEV), which estimates the density ratio in the feature space and offers lower variance. [16] proposed a novel Reverse Validation approach (RV) that leveraged reversed source risk for selection. However, source-based validation methods often necessitate additional model training to handle domain shifts, rendering them cumbersome and less reliable. In contrast, recent model selection methods have shifted their focus exclusively to

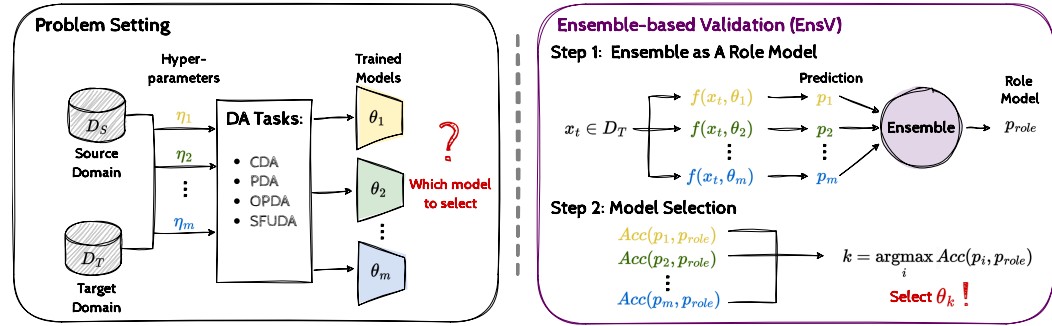

Figure 1: **Left**: Depiction of the unsupervised model selection problem in domain adaptation scenarios, where the objective is to identify the optimal model for the unlabeled target domain. **Right**: Overview of our approach, EnsV, for model selection, which relies solely on predictions of target data by all candidate models.

unlabeled target data, employing specifically designed metrics for model selection. For instance, [17] introduced the mean Shannon's Entropy of target predictions as a model selection metric, promoting confident predictions. [18] proposed the use of Input-Output Mutual Information Maximization (InfoMax)[39] as a metric, augmented with class-balance regularization over Entropy. [19] introduced Soft Neighborhood Density (SND), a novel metric focusing on neighborhood consistency. [20] presented Corr-C, a class correlation-based metric that evaluates both class diversity and prediction certainty simultaneously. Our EnsV baseline aligns with the latter line of research. Importantly, it operates without making any assumptions about cross-domain distribution shifts or the learned target-domain structure, making it suitable for a variety of UDA scenarios. A comprehensive comparison, as presented in Table 2, underscores that EnsV stands out as a simple and versatile approach.

**Ensemble** methods, which harness the collective power of a pool of models through prediction averaging, have been extensively studied in the machine learning community for enhancing model performance [40–43] and improving model calibration [44, 45]. In the era of deep learning, the efficiency of ensembling has garnered significant attention due to the high training cost of deep models. Efficient solutions have been proposed, such as using partially shared parameters [46–48] and leveraging intermediate snapshots [49–51]. Recently, weight averaging has gained attention as an efficient alternative to prediction averaging during inference [52–56]. In addition, diversity is considered crucial for effective ensembles. Various approaches have been explored to achieve diverse checkpoints, including bootstrapping [57], random initializations [58], tuning hyperparameters [59, 60, 53], and combining multiple strategies [61]. Different from mainstream ensemble applications, our work innovatively and elegantly applies ensemble to help address the open problem of unsupervised model selection in various domain adaptation scenarios. In addition, [62] leverages ensembles for hyperparameter selection in CDA but directly uses prediction-based ensembling as the output, unlike our EnsV, which includes a selection step.

## 3 Methodology

We consider a $C$-way image classification task to introduce the concept of unsupervised domain adaptation (UDA). In UDA, we typically have a labeled source domain $\mathcal{D}_\mathrm{s} = \left\{(x_\mathrm{s}^i, y_\mathrm{s}^i)\right\}_{i=1}^{n_\mathrm{s}}$ comprising $n_\mathrm{s}$ annotated source images $x_\mathrm{s}$ and their corresponding labels $y_\mathrm{s}$. Additionally, there is an unlabeled target domain, $\mathcal{D}_\mathrm{t} = \left\{x_\mathrm{t}^i\right\}_{i=1}^{n_\mathrm{t}}$, containing only $n_\mathrm{t}$ unlabeled target images $x_\mathrm{t}$. Despite the tasks being similar, there exist data distribution shifts between the two domains. The primary objective of UDA is to accurately predict the unavailable target labels, $\left\{y_\mathrm{t}^i\right\}_{i=1}^{n_\mathrm{t}}$, by leveraging a discriminative mapping $f(x, \theta)$, which is learned using data from two domains. Here, $\theta \in \mathbb{R}^d$ represents the parameter weights of the trained UDA model. When presented with an input image $x$, the model generates a probability prediction vector, $p = f(x, \theta)$, where $p \in \mathbb{R}^C$ and $\sum_{i=1}^{C} p^i = 1$.

Model selection in UDA is essentially equivalent to the hyperparameter selection challenge. Here, we aim to determine the optimal value for the hyperparameter $\eta$ from a set of $m$ candidate values $\{\eta_i\}_{i=1}^{m}$. The hyperparameter $\eta$ can encompass various aspects, including the learning rate, loss

coefficients, architectural settings, training iterations, and more. By training UDA models using the $m$ different values of $\eta$, we obtain corresponding models with weights denoted as $\{\theta_i\}_{i=1}^m$. In UDA, the objective of model selection is to pinpoint the model $\theta_k$ that demonstrates the best performance on the unlabeled target domain. Subsequently, we select the corresponding hyperparameter $\eta_k$ as the optimal choice for potential adaptation with unlabeled target samples from the exact target domain. We illustrate the problem setting in Figure 1. Without loss of generality, in this paper, we assume $m$ is greater than 1, and candidate models have different weights $\theta$, resulting in different discriminative mappings of $f(x, \theta)$. For clarity, we treat both $\theta$ and the model interchangeably in the presentation. This also applies to model selection, hyperparameter selection, and validation.

## 3.1 Ensemble: The Overlooked 'Free Lunch' in Model Selection

Model selection in UDA is challenging due to the absence of labeled target data for directly evaluating candidate models. Existing selection approaches typically address this challenge from two perspectives: leveraging labeled source data [15] or designing unsupervised metrics based on specific assumed priors [19]. Surprisingly, we've observed that all existing model selection methods treat each candidate model independently, overlooking the collective potential offered by the off-the-shelf ensemble created by these candidates. In this paper, unless otherwise specified, the ensemble refers to prediction-based ensembling, which involves averaging probability predictions across all models to obtain the averaged prediction, i.e., $\frac{1}{m} \sum_{i=1}^m f(x, \theta_i)$ for a sample $x$.

In contrast, we first investigate the potential of the ensemble within the model selection problem. When contemplating the use of the ensemble, two primary concerns often arise, one concerning low efficiency due to training multiple models and the other related to the potential lack of diversity among candidate models. Upon closer inspection of model selection, we observe that the problem setting inherently offers a range of pre-existing candidate models, effectively addressing the efficiency concern without requiring extra model training. Furthermore, all candidate models are trained using a UDA method with varying hyperparameter values, resulting in diverse yet effective discriminative abilities. This naturally mitigates the diversity concern. *Interestingly, the ensemble emerges as a 'free lunch' in UDA model selection, a previously overlooked insight.* To delve deeper into the effectiveness of the ensemble, we present a theoretical analysis grounded in the proposition below.

**Proposition 1** *Given negative log-likelihood (NLL) as the loss function, defined as $l(p, y) = -\log p^y$, and considering a random sample $x$ with label $y$, the following inequality can be established between the loss of the ensemble $\frac{1}{m} \sum_{i=1}^m f(x, \theta_i)$, the averaged loss of all models $\{\theta_i\}_{i=1}^m$, and the loss of the worst one $\theta_{\mathrm{worst}}$:*

$$l(\frac{1}{m} \sum_{i=1}^m f(x, \theta_i), y) < \frac{1}{m} \sum_{i=1}^m l(f(x, \theta_i), y) < l(f(x, \theta_{\mathrm{worst}}), y).$$

Kindly refer to the Appendix for the proof. This proposition theoretically guarantees that the ensemble strictly outperforms the worst candidate model.

## 3.2 Ensemble-based Validation (EnsV): Ensemble as A Role Model for Model Selection

Intuitively, we employ the previously mentioned off-the-shelf ensemble as a reliable role model and select the model that generates predictions closest to this role model among all candidates. To begin with, for each unlabeled target sample $x$, we consider the ensemble $\frac{1}{m} \sum_{i=1}^m f(x, \theta_i)$ as a reliable estimation of its unavailable ground truth. This enables us to obtain reliable predictions for all target data, denoted as $\{\frac{1}{m} \sum_{i=1}^m f(x_j, \theta_i)\}_{j=1}^{n_\mathrm{t}}$. These ensembles can be viewed as the output of a reliable role model, aiding in accurate model selection in the subsequent step. We then utilize the role model to assess all candidate models and select the one with the highest similarity. For simplicity, EnsV involves direct measurement of accuracy between the role model output $\{\frac{1}{m} \sum_{i=1}^m f(x_j, \theta_i)\}_{j=1}^{n_\mathrm{t}}$ and the predictions made by each candidate model, such as $\{f(x_j, \theta_i)\}_{j=1}^{n_\mathrm{t}}$ for the model with weights $\theta_i$. We select the model $\theta_k$ with the highest accuracy and determine the optimal value $\eta_k$ for the hyperparameter $\eta$. Figure 1 provides a vivid illustration of our approach, EnsV. Guided by a reliable role model, EnsV can safely avoid selecting the worst candidate model, a distinct advantage over all existing model selection approaches.

Table 3: Validation accuracy (%) of CDA on *Office-Home* (*Home*). **bold**: Best value.

| Method | ATDOC [35] | | | | | BNM [8] | | | | | CDAN [6] | | | | |
|---|---|---|---|---|---|---|---|---|---|---|---|---|---|---|---|
| | →Ar | →Cl | →Pr | →Re | avg | →Ar | →Cl | →Pr | →Re | avg | →Ar | →Cl | →Pr | →Re | avg |
| SourceRisk [9] | 66.63 | 52.54 | 78.57 | 76.61 | 68.59 | 62.44 | 50.74 | 77.53 | 74.76 | 66.37 | 55.00 | 42.65 | 69.50 | 68.81 | 58.99 |
| IWCV [14] | 67.97 | 54.03 | 78.31 | 79.26 | 69.89 | 66.56 | 48.16 | 74.09 | 73.28 | 65.52 | 61.31 | 41.24 | 67.17 | 71.93 | 60.41 |
| DEV [15] | 67.39 | 54.23 | 77.78 | 79.39 | 69.70 | 65.76 | 56.39 | 73.92 | 77.59 | 68.41 | 67.23 | 57.04 | 68.76 | 76.91 | 67.49 |
| RV [16] | 68.68 | 56.13 | 78.93 | 79.64 | 70.85 | 68.25 | 56.75 | **78.08** | 78.67 | 70.44 | 67.66 | 56.74 | 76.01 | 77.68 | 69.52 |
| Entropy [17] | 63.67 | 55.83 | 76.54 | 78.36 | 68.60 | 66.28 | 54.49 | 74.15 | 77.64 | 68.14 | 67.66 | **57.56** | 76.37 | 77.45 | 69.76 |
| InfoMax [18] | 63.67 | 55.63 | 77.61 | 78.36 | 68.82 | 66.28 | 54.49 | 74.15 | 77.64 | 68.14 | 67.66 | **57.56** | 76.37 | 77.45 | 69.76 |
| SND [19] | 63.67 | 55.63 | 76.54 | 77.54 | 68.34 | 66.28 | 54.49 | 74.15 | 77.64 | 68.14 | 67.94 | **57.56** | 76.96 | 77.68 | 70.04 |
| Corr-C [20] | 63.51 | 50.39 | 73.89 | 73.88 | 65.42 | 58.10 | 45.37 | 68.97 | 70.59 | 60.76 | 53.84 | 41.21 | 64.96 | 67.65 | 56.91 |
| EnsV | **68.70** | **58.05** | **79.81** | **80.41** | **71.74** | **68.61** | **57.38** | **78.08** | **79.54** | **70.90** | **67.88** | **57.56** | **77.39** | **78.19** | **70.25** |
| Worst | 62.89 | 50.39 | 73.89 | 73.88 | 65.26 | 58.10 | 45.37 | 68.96 | 70.59 | 60.75 | 53.80 | 41.21 | 64.78 | 67.65 | 56.86 |
| Best | 68.97 | 58.35 | 80.27 | 80.58 | 72.04 | 68.93 | 57.51 | 78.43 | 79.57 | 71.11 | 68.19 | 57.90 | 77.44 | 78.19 | 70.43 |

| Method | MCC [36] | | | | | MDD [33] | | | | | SAFN [7] | | | | | *Home* |
|---|---|---|---|---|---|---|---|---|---|---|---|---|---|---|---|---|
| | →Ar | →Cl | →Pr | →Re | avg | →Ar | →Cl | →Pr | →Re | avg | →Ar | →Cl | →Pr | →Re | avg | AVG |
| SourceRisk [9] | 66.57 | 56.53 | 79.55 | 80.90 | 70.89 | 62.53 | 54.43 | 75.27 | 75.55 | 66.94 | 63.54 | 51.34 | 73.66 | 74.54 | 65.77 | 66.26 |
| IWCV [14] | 68.69 | 58.93 | 80.37 | 80.08 | 72.02 | 64.20 | 56.50 | 73.78 | 74.28 | 67.19 | 64.31 | **52.36** | 72.31 | 74.29 | 65.82 | 66.81 |
| DEV [15] | 68.81 | 58.07 | 78.54 | 80.10 | 71.38 | 64.42 | 56.94 | 76.85 | 75.94 | 68.54 | 63.15 | 50.47 | 71.20 | 74.54 | 64.84 | 68.39 |
| RV [16] | **70.40** | 58.80 | **80.63** | 80.39 | 72.56 | **66.57** | 55.75 | 76.60 | 76.90 | 68.96 | 64.31 | 50.13 | 73.77 | 74.93 | 65.78 | 69.68 |
| Entropy [17] | 69.29 | 59.33 | **80.63** | 80.96 | 72.55 | 66.54 | 57.63 | 77.27 | **77.45** | 69.72 | 59.85 | 46.41 | 72.51 | 73.18 | 62.99 | 68.63 |
| InfoMax [18] | 66.58 | 58.48 | 79.12 | 80.81 | 71.25 | 66.54 | 57.74 | 77.27 | **77.45** | 69.75 | 64.56 | 49.71 | 73.77 | 73.18 | 65.31 | 68.84 |
| SND [19] | 69.05 | 55.61 | 79.72 | 79.10 | 70.87 | 51.34 | 38.01 | **77.61** | 68.46 | 58.86 | 57.90 | 46.41 | 67.04 | 68.18 | 59.88 | 66.02 |
| Corr-C [20] | 69.05 | 55.61 | 79.72 | 79.10 | 70.87 | 47.79 | 31.69 | 63.40 | 60.63 | 50.88 | 62.66 | 46.41 | 66.83 | 68.18 | 61.52 | 61.06 |
| EnsV | 69.92 | **59.50** | 80.30 | 80.86 | **72.65** | 66.46 | **57.81** | **77.61** | 76.51 | 69.60 | **65.91** | 52.18 | **74.51** | **75.57** | **67.04** | **70.36** |
| Worst | 62.72 | 54.63 | 76.19 | 78.19 | 67.93 | 47.79 | 31.69 | 63.40 | 60.63 | 50.88 | 57.90 | 46.41 | 67.04 | 68.18 | 59.88 | 60.26 |
| Best | 70.68 | 59.95 | 80.93 | 81.02 | 73.14 | 66.75 | 58.36 | 77.61 | 77.45 | 70.04 | 66.59 | 53.14 | 74.90 | 75.57 | 67.55 | 70.72 |

# 4 Experiments

## 4.1 Setup

**Datasets** Our experiments encompass diverse and widely-used image classification benchmarks: (*i*) *Office-31* [63] with 31 classes and 3 domains (Amazon (A), DSLR (D), and Webcam (W)); (*ii*) *Office-Home* [64] with 65 classes and 4 domains (Art (Ar), Clipart (Cl), Product (Pr), and Real-World (Re)); (*iii*) *VisDA* [65] with 12 classes and 2 domains (training (T) and validation (V)); and (*iv*) *DomainNet-126* [13, 5] with 126 classes and 4 domains (Real (R), Clipart (C), Painting (P), and Sketch (S)). Additionally, we conduct experiments in synthetic-to-real semantic segmentation, specifically targeting the transfer from *GTAV* [66] to *Cityscapes* [67].

**UDA methods** In our experiments, we assess all the model selection approaches listed in Table 2. Kindly refer to the Appendix for detailed introductions of them. With these approaches, we perform model selection for various UDA methods across different UDA settings. For CDA of image classification, we consider ATDOC [35], BNM [8], CDAN [6], MCC [36], MDD [33], and SAFN [7]. For PDA, we consider PADA [10] and SAFN [7]. For OPDA, we consider DANCE [11]. For SFUDA, we consider the white-box method SHOT [12] and the black-box method DINE [22]. For domain adaptive semantic segmentation, we consider AdaptSeg [25] and AdvEnt [26]. Following previous model selection studies [15, 19], we primarily focus on one-hyperparameter validation and present the comprehensive hyperparameter settings for all UDA methods in the Appendix. For each hyperparameter, we generally explore 7 candidate values. Additionally, we perform two types of challenging two-hyperparameter validation tasks. For classification tasks, we select the bottleneck dimension as the second hyperparameter from 4 options: 256, 512, 1024, 2048 in MCC and MDD. For segmentation tasks, following SND [19], we select the training iteration as the second hyperparameter from 8 options, ranging from 16,000 to 30,000 iterations at intervals of 2,000 iterations, in AdaptSeg and AdvEnt.

**Implementation details** For all UDA methods, we train UDA models using the Transfer Learning Library* or the official GitHub code on a single RTX TITAN 16GB GPU with a batch size of 32 and a total number of iterations of 5000. Unless specified, checkpoints are saved at the last iteration. We adopt ResNet-101 [68] for *VisDA* and segmentation tasks, ResNet-34 [68] for *DomainNet*, and ResNet-50 [68] for other benchmarks. We assess the selection performance of all model selection methods on our trained models for fair comparisons. As a result, comparing our reported values with those from the original papers [15, 19] would be inappropriate. We repeat trials with three random seeds and report the mean for results. Source-based validation methods allocate 80% of the source data for training and the remaining 20% for validation.

---

*https://github.com/thuml/Transfer-Learning-Library

Table 4: Validation accuracy (%) of CDA on *Office-31* (*Office*) and *VisDA*.

| Method | ATDOC [35] | | | | | BNM [8] | | | | | CDAN [6] | | | | |
|---|---|---|---|---|---|---|---|---|---|---|---|---|---|---|---|
| | →A | →D | →W | avg | T→V | →A | →D | →W | avg | T→V | →A | →D | →W | avg | T→V |
| SourceRisk [9] | 72.56 | 88.96 | **87.80** | 83.11 | 67.79 | 72.92 | **90.36** | 89.43 | 84.24 | 70.51 | 63.90 | 91.16 | **89.06** | 81.37 | 64.50 |
| IWCV [14] | 72.56 | 86.14 | 86.54 | 81.75 | 67.79 | 72.92 | 85.54 | 89.43 | 82.63 | 76.94 | 63.90 | 69.08 | 58.74 | 63.91 | 64.50 |
| DEV [15] | 72.56 | 86.14 | 86.54 | 81.75 | 70.34 | 72.92 | 85.54 | 89.43 | 82.63 | 64.50 | 63.90 | 91.16 | 88.30 | 81.12 | 64.50 |
| RV [16] | **74.93** | 89.96 | 87.23 | 84.04 | **77.37** | 70.71 | 88.55 | 89.43 | 82.90 | 74.58 | **73.27** | 91.16 | 88.30 | 84.24 | 76.02 |
| Entropy [17] | 73.29 | 86.14 | **87.80** | 82.41 | 62.85 | 72.67 | 85.54 | 83.14 | 80.45 | 58.36 | 71.62 | 91.16 | **89.06** | 83.95 | **80.46** |
| InfoMax [18] | 73.29 | 86.14 | **87.80** | 82.41 | 76.49 | 70.52 | 85.54 | 83.14 | 79.73 | 58.36 | 71.62 | 91.16 | 88.30 | 83.69 | **80.46** |
| SND [19] | 73.29 | **92.37** | **87.80** | 84.49 | 77.37 | 74.44 | 85.54 | 83.14 | 81.04 | 69.65 | 71.55 | 92.37 | 88.55 | 84.16 | **80.46** |
| Corr-C [20] | 71.05 | 90.96 | 84.40 | 82.14 | 67.79 | 67.16 | 84.34 | 78.99 | 76.83 | 70.51 | 58.29 | 67.67 | 59.62 | 61.86 | 64.50 |
| EnsV | 74.83 | 90.96 | **87.80** | **84.53** | 73.36 | **74.87** | **90.36** | 89.43 | **84.89** | 74.58 | 73.20 | **92.77** | 88.55 | **84.84** | 79.05 |
| Worst | 71.05 | 86.14 | 84.40 | 80.53 | 62.85 | 67.16 | 84.34 | 78.99 | 76.83 | 23.08 | 58.29 | 67.67 | 57.11 | 61.02 | 64.50 |
| Best | 75.31 | 92.37 | 87.80 | 85.16 | 77.37 | 75.52 | 90.36 | 89.43 | 85.10 | 76.94 | 73.38 | 92.77 | 89.06 | 85.07 | 80.46 |

| Method | MCC [36] | | | | | MDD [33] | | | | | SAFN [7] | | | | | *Office* | *VisDA* |
|---|---|---|---|---|---|---|---|---|---|---|---|---|---|---|---|---|---|
| | →A | →D | →W | avg | T→V | →A | →D | →W | avg | T→V | →A | →D | →W | avg | T→V | AVG | AVG |
| SourceRisk [9] | 73.11 | 90.96 | 91.07 | 85.05 | 80.46 | 75.72 | 91.06 | 86.23 | 84.34 | 72.25 | 69.20 | 83.73 | 87.17 | 80.03 | 70.71 | 83.02 | 71.04 |
| IWCV [14] | 73.11 | 91.16 | 88.55 | 84.27 | 81.48 | 75.49 | 91.16 | 89.18 | 85.28 | 72.25 | 69.32 | 86.55 | 80.38 | 78.75 | 66.33 | 79.43 | 71.55 |
| DEV [15] | 72.70 | 89.16 | 93.08 | 84.98 | 81.48 | 75.65 | 91.16 | 89.18 | 85.33 | 72.25 | 68.21 | 86.55 | 80.38 | 78.38 | 66.33 | 82.36 | 71.97 |
| RV [16] | **73.97** | 89.06 | 93.08 | 85.37 | **82.22** | 74.46 | **92.57** | 86.79 | 84.61 | 77.23 | 68.69 | 90.83 | 87.17 | 82.23 | 66.33 | 83.90 | 75.62 |
| Entropy [17] | 73.93 | 90.56 | **93.46** | 85.98 | **82.22** | 76.31 | **92.57** | 90.82 | 86.57 | 78.95 | 68.23 | **91.57** | 85.66 | 81.82 | 70.20 | 83.53 | 72.17 |
| InfoMax [18] | 73.93 | 89.16 | 88.55 | 83.88 | 81.48 | **76.50** | **92.57** | 90.82 | **86.63** | **78.95** | 68.23 | **91.57** | 87.42 | 82.41 | 70.20 | 83.13 | 74.32 |
| SND [19] | 73.93 | **91.97** | **93.46** | **86.45** | 69.35 | **76.50** | 92.17 | 90.82 | 86.50 | **78.95** | 68.23 | 89.96 | 85.66 | 81.28 | 58.15 | 83.99 | 72.32 |
| Corr-C [20] | 73.93 | 91.37 | **93.46** | 86.25 | 69.35 | 74.25 | 91.57 | 85.66 | 83.83 | 72.25 | 68.39 | 86.75 | 80.38 | 78.51 | 62.52 | 78.24 | 67.82 |
| EnsV | 73.75 | 90.56 | 91.45 | 85.25 | **82.22** | 75.92 | 92.57 | 90.82 | 86.44 | 77.23 | 69.67 | 90.96 | 87.17 | **82.60** | **73.96** | **84.76** | **76.73** |
| Worst | 70.56 | 86.75 | 87.17 | 81.49 | 69.35 | 73.06 | 87.35 | 85.66 | 82.02 | 72.25 | 67.27 | 83.73 | 80.38 | 77.13 | 58.15 | 76.50 | 58.36 |
| Best | 74.42 | 91.97 | 93.46 | 86.62 | 82.23 | 76.52 | 92.57 | 92.20 | 87.10 | 78.95 | 70.06 | 91.57 | 87.42 | 83.02 | 75.30 | 85.34 | 78.54 |

Table 5: Validation accuracy (%) of PDA on *Office-Home*.

| Method | SAFN [7] | | | | | PADA [10] | | | | | *Home* |
|---|---|---|---|---|---|---|---|---|---|---|---|
| | → Ar | → Cl | → Pr | → Re | avg | → Ar | → Cl | → Pr | → Re | avg | AVG |
| SourceRisk [9] | 66.82 | 54.71 | 74.41 | 76.48 | 68.11 | 57.21 | 41.90 | 64.48 | 71.89 | 58.87 | 63.49 |
| IWCV [14] | 69.36 | 53.91 | 71.78 | 76.38 | 67.86 | 59.65 | 50.51 | 66.84 | 72.96 | 62.49 | 65.18 |
| DEV [15] | 69.36 | 54.94 | 73.95 | 76.06 | 68.58 | 66.88 | 49.29 | 72.40 | 70.46 | 64.76 | 66.67 |
| RV [16] | 68.98 | 52.74 | 72.83 | 77.14 | 67.92 | 57.79 | 40.87 | 63.87 | 70.83 | 58.34 | 63.13 |
| Entropy [17] | **71.75** | 55.62 | 76.36 | 76.59 | 70.08 | 60.08 | 46.51 | 53.16 | 62.47 | 55.56 | 62.82 |
| InfoMax [18] | 63.67 | 51.74 | 69.64 | 73.62 | 64.67 | 60.08 | 51.40 | 60.20 | 66.67 | 59.59 | 62.13 |
| SND [19] | **71.75** | 51.74 | 76.36 | 78.36 | 69.55 | 67.80 | 50.71 | 59.46 | 67.13 | 61.27 | 65.41 |
| Corr-C [20] | 71.23 | 55.70 | **76.94** | **79.13** | **70.75** | 61.34 | 45.65 | 54.90 | 62.25 | 56.04 | 63.40 |
| EnsV | 70.98 | **56.12** | 75.67 | 78.48 | 70.31 | **68.54** | **55.60** | **69.86** | **78.23** | **68.06** | **69.19** |
| Worst | 62.48 | 49.91 | 68.50 | 73.62 | 63.63 | 56.29 | 39.76 | 50.49 | 59.31 | 51.46 | 57.55 |
| Best | 73.37 | 58.09 | 77.35 | 79.33 | 72.03 | 69.33 | 55.86 | 74.55 | 79.59 | 69.83 | 70.93 |

## 4.2 Comprehensive Comparison of All Model Selection Methods

Following prior studies [15, 19, 18], we extensively compare our EnsV with 8 other methods in standard UDA settings, including CDA and PDA. Averaged results are presented for UDA tasks sharing the same target domain. For example, results of 'Cl→Ar', 'Pr→Ar', and 'Re→Ar' on *Office-Home* are averaged and reported under the column labeled '→ Ar'. In addition, the column 'avg' signifies the averaged results for each UDA method while the 'AVG' row represents the averaged results across different UDA methods. 'Worst' refers to the worst candidate model with the lowest target-domain performance, while 'Best' indicates the best candidate model with the highest performance. Kindly refer to the Appendix for full results.

**CDA** We provide model selection results for 6 typical closed-set UDA methods on *Office-Home*, *Office-31*, and *VisDA* in Tables 3 and 4. EnsV consistently outperforms other validation methods in terms of the average selection accuracy on each benchmark and consistently achieves near-best model selection results. Among existing methods, we find the reverse validation (RV) approach is consistently the best among the three benchmarks. However, RV requires extra model re-training, making it impractical when compared to the efficient target-specific model selection methods.

**PDA** For partial-set UDA with label shift of missing source-domain classes, we conduct hyper-parameter selections for two different UDA methods on *Office-Home* (Table 5). Notably, existing methods, except for DEV and SND, suffer from frequent low-accuracy selections. In contrast, EnsV consistently achieves high-accuracy selections and, on average, outperforms both DEV and SND.

## 4.3 Comparison of Target-specific Model Selection Methods

Recent advancements in UDA model selection [19, 18] indicate that validation using only unlabeled target data can achieve superior performance compared to source-based methods, with increased simplicity. Eliminating the reliance on source data facilitates easy application in various real-world UDA scenarios, extending beyond conventional closed-set settings. We particularly compare EnsV

Table 6: Validation accuracy (%) of CDA on *DomainNet-126 (DNet)*.

| Method | CDAN [6] | | | | | BNM [8] | | | | | ATDOC [35] | | | | | *DNet* AVG |
|---|---|---|---|---|---|---|---|---|---|---|---|---|---|---|---|---|
| | →C | →P | →R | →S | avg | →C | →P | →R | →S | avg | →C | →P | →R | →S | avg | |
| Entropy [17] | **67.09** | **65.80** | 74.42 | **59.34** | **66.66** | 63.36 | 64.28 | 74.31 | 48.69 | 62.66 | 63.75 | 61.85 | 79.60 | 52.17 | 64.34 | 64.55 |
| InfoMax [18] | **67.09** | **65.80** | 74.42 | **59.34** | **66.66** | 67.05 | 64.28 | 74.31 | 55.67 | 65.33 | 63.75 | 61.85 | 79.60 | 52.17 | 64.34 | 65.44 |
| SND [19] | **67.09** | 64.68 | 74.42 | **59.34** | 66.38 | 56.56 | 54.50 | 74.31 | 42.37 | 56.93 | 63.75 | 61.85 | 79.60 | 47.00 | 63.05 | 62.12 |
| Corr-C [20] | 57.35 | 62.88 | 74.42 | 54.63 | 62.32 | 59.75 | 63.41 | **77.62** | 42.37 | 60.79 | 59.98 | 62.27 | 74.42 | 53.69 | 62.59 | 61.90 |
| EnsV | 65.88 | 65.27 | **74.44** | 57.42 | 65.75 | **67.86** | **66.06** | **77.62** | **57.69** | **67.31** | **70.30** | **68.44** | **80.01** | **61.73** | **70.12** | **67.73** |
| Worst | 57.35 | 60.76 | 73.44 | 51.41 | 60.74 | 55.79 | 54.50 | 74.31 | 42.37 | 56.74 | 59.98 | 61.85 | 74.42 | 47.00 | 60.81 | 59.43 |
| Best | 67.09 | 65.80 | 74.44 | 59.34 | 66.66 | 67.86 | 66.50 | 78.68 | 58.49 | 67.88 | 70.30 | 68.44 | 80.38 | 62.23 | 70.34 | 68.29 |

with other target-specific validation methods on the large-scale benchmark *DomainNet-126* and in two extra practical UDA settings: OPDA and SFUDA.

**CDA** We compare all target-specific validation methods on the large-scale benchmark *DomainNet-126* (Table 6). EnsV consistently keeps the leading validation performance, while other approaches exhibit high variance.

Table 7: H-score [69, 70] (%) of an OPDA method DANCE [11] on *Office-Home*.

| Method | Ar → Cl | Ar → Pr | Ar → Re | Cl → Ar | Cl → Pr | Cl → Re | Pr → Ar | Pr → Cl | Pr → Re | Re → Ar | Re → Cl | Re → Pr | avg |
|---|---|---|---|---|---|---|---|---|---|---|---|---|---|
| Entropy [17] | 38.29 | 26.08 | 36.51 | 32.92 | 17.10 | 32.19 | 37.69 | 46.40 | 45.53 | 25.39 | 33.75 | 39.37 | 34.27 |
| InfoMax [18] | 38.29 | 26.08 | 36.51 | 32.92 | 17.10 | 32.19 | 37.69 | 46.40 | 45.33 | 25.39 | 33.75 | 39.37 | 34.25 |
| SND [19] | 1.00 | 0.00 | 12.73 | 0.00 | 42.84 | 1.95 | 19.77 | 11.99 | 35.69 | 25.39 | 0.00 | 28.40 | 14.98 |
| Corr-C [20] | 1.00 | 0.00 | 12.73 | 0.00 | 42.84 | 1.95 | 19.77 | 11.99 | 35.69 | 69.02 | 0.00 | 28.40 | 18.62 |
| EnsV | **38.40** | **76.96** | **66.57** | **71.76** | **75.17** | **69.99** | **77.42** | **48.15** | **69.40** | **81.84** | **67.54** | **84.31** | **68.96** |
| Worst | 1.00 | 0.00 | 12.73 | 0.00 | 17.10 | 1.95 | 19.77 | 11.99 | 35.69 | 25.39 | 0.00 | 28.40 | 12.84 |
| Best | 67.00 | 76.96 | 66.57 | 71.76 | 75.17 | 69.99 | 77.42 | 64.32 | 72.87 | 81.84 | 67.54 | 84.31 | 72.98 |

**OPDA** In open-partial-set UDA with label shift of unknown classes, we choose a representative method DANCE for validation on *Office-Home* (Table 7) and measure the H-score [70, 69]. Previous validation works have not studied this challenging setting [19], and all of them encounter issues with poor model selections. In contrast, EnsV consistently achieves high-accuracy selections.

Table 8: Validation accuracy (%) of SFUDA on *Office-Home*, *Office-31*, and *VisDA*.

| Method | SHOT [12] | | | | | SHOT [12] | | | | DINE [22] |
|---|---|---|---|---|---|---|---|---|---|---|
| | →Ar | →Cl | →Pr | →Re | avg | →A | →D | →W | avg | T→V |
| Entropy [17] | 63.38 | 50.45 | 77.35 | 77.65 | 67.21 | 71.67 | 90.76 | 88.68 | 83.70 | 71.99 |
| InfoMax [18] | 63.38 | 50.45 | 77.35 | 77.65 | 67.21 | 71.67 | 90.76 | 88.68 | 83.70 | 71.99 |
| SND [19] | 64.58 | 54.17 | 78.23 | 77.65 | 68.66 | 71.67 | 90.76 | 88.68 | 83.70 | **74.43** |
| Corr-C [20] | 69.13 | 56.32 | 79.29 | 79.14 | 70.97 | 71.58 | 90.76 | 90.19 | 84.18 | 71.99 |
| EnsV | **69.58** | **56.78** | **80.40** | **80.76** | **71.88** | **74.85** | **94.78** | **91.82** | **87.15** | **74.43** |
| Worst | 63.38 | 50.45 | 77.35 | 77.65 | 67.21 | 71.56 | 90.76 | 88.68 | 83.67 | 71.99 |
| Best | 69.83 | 57.08 | 80.55 | 80.76 | 72.05 | 75.06 | 94.78 | 93.33 | 87.72 | 76.17 |

**SFUDA** In source-free UDA, where source-based model selection methods are not applicable due to no access to source data, we select SHOT for the white-box setting on *Office-31* and DINE for the black-box setting on *VisDA* (Table 8). EnsV consistently maintains near-best selections, while other target-based approaches frequently make worst-case selections.

Table 9: Worst-case selections of various target domain-specific model selection approaches, which are reported as the H-score (%) for OPDA and accuracy (%) for other UDA settings.

| Method | CDA | | | | | | PDA | | OPDA | | SFUDA | |
|---|---|---|---|---|---|---|---|---|---|---|---|---|
| | ATDOC | ATDOC | BNM | BNM | MDD | SAFN | PADA | PADA | DANCE | DANCE | SHOT | DINE |
| | Cl→Ar | C→S | Ar→Pr | R→S | Pr→Cl | Pr→Cl | Ar→Re | Re→Ar | Re→Ar | Pr→Re | D→A | T→V |
| Entropy [17] | 59.25 | 46.43 | 67.04 | 40.95 | 55.85 | 43.30 | 55.94 | 70.52 | 25.39 | 45.53 | 71.21 | 71.99 |
| InfoMax [18] | 59.25 | 46.43 | 67.04 | 54.93 | 55.85 | 43.30 | 78.02 | 70.52 | 25.39 | 45.53 | 71.21 | 71.99 |
| SND [19] | 59.25 | 46.43 | 67.04 | 40.95 | 21.60 | 43.30 | 55.94 | 74.66 | 25.39 | 35.69 | 71.21 | 74.43 |
| Corr-C [20] | 59.37 | 54.71 | 67.06 | 40.95 | 21.60 | 43.30 | 55.94 | 71.26 | 69.02 | 35.69 | 71.21 | 71.99 |
| EnsV | **66.25** | **62.11** | **77.00** | **57.65** | **57.02** | **49.69** | **86.53** | **76.86** | **81.84** | **69.40** | **75.15** | **74.43** |
| Worst | 59.25 | 46.43 | 67.04 | 40.95 | 21.60 | 43.30 | 55.94 | 70.52 | 25.39 | 35.69 | 71.21 | 71.99 |
| Best | 66.91 | 63.12 | 77.00 | 58.50 | 57.02 | 50.52 | 86.53 | 76.86 | 81.84 | 72.87 | 75.15 | 76.17 |

**Worst-model selection comparisons** For empirical evidence of the superiority of EnsV, we compare EnsV with other target-specific methods, specifically focusing on worst-case avoidance, through specific examples presented in Table 9. In short, EnsV consistently avoids the worst selections, while other methods often encounter significant challenges.

Table 10: CDA accuracy (%) on *Office-Home* when two hyperparameters are validated.

| Method | MDD [33] | | | | | MCC [36] | | | | | |
|---|---|---|---|---|---|---|---|---|---|---|---|
| | Ar → Cl | Cl → Pr | Pr → Re | Re → Ar | avg | Ar → Cl | Cl → Pr | Pr → Re | Re → Ar | avg | AVG |
| SourceRisk | 55.99 | 73.15 | 78.77 | 69.39 | 69.33 | 57.91 | 76.84 | 81.13 | 72.89 | 72.19 | 70.76 |
| IWCV [14] | 37.89 | 72.92 | 80.42 | 58.43 | 62.42 | 46.09 | 77.74 | 80.68 | 74.45 | 69.74 | 66.08 |
| DEV [15] | 52.60 | 72.11 | 53.36 | 67.70 | 61.44 | 59.47 | 76.84 | 81.94 | 74.08 | 73.08 | 67.26 |
| RV [16] | 57.59 | 72.25 | 80.83 | 70.79 | 70.37 | 59.13 | 76.84 | 82.03 | 71.98 | 72.50 | 71.44 |
| Entropy [17] | 57.21 | **73.19** | 80.06 | **72.31** | 70.69 | 59.75 | 77.77 | 82.37 | 74.33 | 73.56 | 72.13 |
| InfoMax [18] | **57.59** | 72.92 | 80.06 | **72.31** | **70.72** | 59.70 | **78.73** | **82.58** | 70.33 | 72.84 | 71.78 |
| SND [19] | 38.10 | 56.45 | 70.03 | 65.10 | 57.42 | 53.49 | 74.97 | 77.25 | 74.12 | 69.96 | 63.69 |
| Corr-C [20] | 30.17 | 44.74 | 57.15 | 50.76 | 45.71 | 44.90 | 56.75 | 74.32 | 67.61 | 60.90 | 53.31 |
| EnsV-P | 56.91 | 72.74 | **80.93** | 71.16 | 70.44 | **60.39** | 78.71 | 82.28 | **74.91** | **74.07** | **72.26** |
| Worst | 30.17 | 39.81 | 53.36 | 50.76 | 43.53 | 43.02 | 56.75 | 73.47 | 67.24 | 60.12 | 51.83 |
| Best | 57.59 | 73.35 | 80.93 | 72.52 | 71.10 | 61.10 | 78.94 | 83.04 | 75.36 | 74.61 | 72.86 |

## 4.4 Further Analysis

**Validation with two hyperparameters** We conduct two-hyperparameters model selection experiments with a large pool of model candidates, i.e., 28 models for image classification (Table 10) and 48 models for image segmentation (Table 11). EnsV consistently achieves near-optimal selections in both scenarios, surpassing other versatile validation methods such as Entropy and SND.

Table 11: Segmentation mIoU (%) of AdaptSeg and AdvEnt on *GTAV → Cityscapes* when two hyperparameters are validated.

| Method | AdaptSeg [25] | AdvEnt [26] |
|---|---|---|
| SourceRisk [9] | 39.52 | 39.08 |
| Entropy [17] | 39.47 | 38.41 |
| SND [19] | **40.69** | 40.02 |
| EnsV | **40.69** | **40.67** |
| Worst | 35.32 | 34.22 |
| Best | 42.20 | 41.78 |

Table 12: CDA accuracy (%) of BNM with ViT as the backbone.

| Method | BNM [8] |
|---|---|
| Entropy [17] | 28.21 |
| InfoMax [18] | 28.21 |
| SND [19] | 52.42 |
| Corr-C [20] | 28.21 |
| EnsV | **55.16** |
| Worst | 28.21 |
| Best | 55.16 |

**Robustness to architectures** In our experiments, we evaluate the robustness of EnsV across various ResNet backbone variants, observing consistent success across different scales. We also conduct validation experiments using the ViT-B architecture [71] on the R→S task with BNM. The validation results, presented in Table 12, demonstrate that EnsV achieves the best selection. However, all other target-based methods except SND make the worst selection.

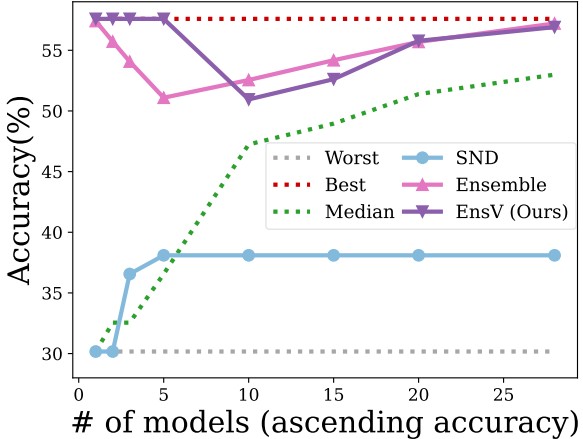

Figure 2: For the 28 candidate models available in the two-hyperparameter selection task with MDD on Ar→Cl, we first rank them based on their respective actual target-domain accuracy. We then start with only the best candidate model in the pool and gradually add increasingly inferior models in ascending order of their accuracy. The figure illustrates how adding more inferior models affects the performance of the ensemble and model selection.

**Robustness to poor candidates** Ensuring the ensemble's resilience to poor models is crucial for its broad effectiveness. We assess this by conducting a two-hyperparameters model selection task for MDD on Ar→Cl. We consider a challenging scenario where only the best candidate model is initially in the pool, gradually adding an increasing number of inferior models. This setup allows us to examine how the ensemble performs when dominated by inferior models. From the results shown in Figure 2, we find that both Ensemble and EnsV still consistently achieve performance above the median, demonstrating their resilience. In contrast, SND [19], a state-of-the-art method, struggles to surpass the median accuracy.

## 5 Discussions

**Limitations** EnsV selects the candidate model that most closely matches the ensemble's performance. While EnsV consistently avoids selecting the worst-performing models, its performance can be suboptimal if the ensemble (role model) itself is suboptimal. EnsV may face challenges in the following scenarios:

- Single high-performing vs. poor model: when the model pool contains only one high-performing model and one poor model, EnsV may struggle. In such cases, the ensemble of just these two models might not accurately reflect the quality of the good model.

- Small performance differences: if the performance differences among all candidate models are small, EnsV may have difficulty distinguishing between them. In such scenarios, using ensemble results for model selection may not provide the fine-grained control needed to accurately differentiate between the candidates.

- Predominantly poor models: when the majority of models in the pool are poor, with only a few having normal or good performance, EnsV may encounter issues. An ensemble composed mostly of poor models may produce results that are closer to the poor models, leading to a final selection of a suboptimal candidate.

**Takeaways** Following a thorough empirical comparison of existing UDA model selection approaches, several key conclusions emerge:

- The significance of model selection in influencing the deployment performance of UDA methods becomes evident. Relying on fixed hyperparameters or limited analyses is inadequate. We emphasize the importance of increased attention and transparent reporting of validation methods, consistent with recommendations in [15, 19, 18].

- Among existing validation methods, we recommend the reverse validation (RV) approach, which, despite being overlooked in previous studies [15, 19, 18], proves to be the most reliable method for widely studied closed-set UDA scenarios when source data is available. However, it requires additional model re-training, making it less lightweight compared to target-based validation methods. Moreover, all existing model selection methods demonstrate unreliability across diverse UDA methodologies and real-world settings such as open-set and source-free UDA. These methods struggle to maintain effectiveness, posing a significant risk to the successful application of UDA in various scenarios.

- Regarding our proposed baseline, EnsV, we believe it is a simple and versatile model selection method that is certified to avoid worst-case selections. While it may not always achieve peak performance, especially when the ensemble result is suboptimal, EnsV offers valuable insights for future explorations in reliable model selection methods.

## Acknowledgements

Jian Liang was funded by the Beijing Nova Program under Grant Z211100002121108, the National Natural Science Foundation of China under Grant 62276256, and the Young Elite Scientists Sponsorship Program by CAST (2023QNRC001).

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

# A  Proof of Proposition 1

We first prove the first inequality using Jensen's inequality, which states that for a real-valued, convex function $\varphi$ with its domain as a subset of $\mathbb{R}$ and numbers $t_1, \ldots, t_n$ in its domain, the inequality $\varphi\left(\frac{1}{n}\sum_{i=1}^{n} t_i\right) \leq \frac{1}{n}\sum_{i=1}^{n} \varphi(t_i)$ holds. Given that $-\log$ is convex, and assuming $m > 1$ with candidate models having different parameter weights $\theta$, resulting in distinct discriminative mappings of $f(x, \theta)$, we can strictly obtain $l\left(\frac{1}{m}\sum_{i=1}^{m} f(x, \theta_i), y\right) < \frac{1}{m}\sum_{i=1}^{m} l(f(x, \theta_i), y)$ without the equal situation. Next, we leverage the property of inequalities to prove the second inequality. Here, $\theta_{\text{worst}}$ denotes the worst candidate model, i.e., the model with the largest loss. For any other candidate model $\theta_i$, we have $l(f(x, \theta_i), y) < l(f(x, \theta_{\text{worst}}), y)$. This ensures that $\frac{1}{m}\sum_{i=1}^{m} l(f(x, \theta_i), y) < \frac{1}{m}\sum_{i=1}^{m} l(f(x, \theta_{\text{worst}}), y)$, or explicitly, $\frac{1}{m}\sum_{i=1}^{m} l(f(x, \theta_i), y) < l(f(x, \theta_{\text{worst}}), y)$. Substituting the NLL loss with any strongly convex loss function would still uphold the proposition.

# B  Model Selection Baselines

Let $\{p_t^i\}_{i=1}^{n_t}$ represent the output probability vectors of all $n_t$ target samples, and let $P \in \mathbb{R}^{n_t \times C}$ denote the total probability matrix. We introduce the respective computation involved in the existing model selection approaches.

**Source risk.**  The SourceRisk approach [9] utilizes a held-out validation set from the source domain to select the model $\theta_k$ that performs best on this set as the final decision. However, this method has limited effectiveness in scenarios with severe domain shifts between the source and target domains. Additionally, it introduces additional hyperparameters for dataset splitting, which can further complicate the model selection process.

**Importance-weighted source risk.**  Directly taking source risk as target risk is unreliable due to domain distribution shifts between domains. To address this challenge, [14] propose Importance-Weighted Cross Validation (IWCV), which re-weights the source risk using a source-target density ratio estimated in the input space. [15] further enhance IWCV by introducing Deep Embedded Validation (DEV), which estimates the density ratio in the feature space using a domain discriminator and controls the variance. Both IWCV and DEV rely on the importance weighting technique [72], which assumes that the target distribution is included in the source distribution [14], making the weighting unreliable in scenarios with severe covariate shift and label shift. In addition, both IWCV and DEV involve hyperparameters and extra model training during the density ratio estimation process.

**Reversed source risk.**  Building upon the concept of reverse cross-validation [73], [16] propose a novel Reverse Validation approach (RV). This method first conducts source-to-target adaptation to obtain a UDA model, which enables the acquisition of pseudo labels for the target unlabeled data. Subsequently, Reverse Validation performs a reversed adaptation from the pseudo-labeled target to the source and utilizes the source risk in this reversed adaptation task for validation. Reverse Validation relies on the symmetry between domains and cannot handle label shifts. Additionally, this approach involves hyperparameters for dataset splitting.

**Entropy.**  [17] propose using the mean Shannon's Entropy of target-domain predictions as a validation metric, prioritizing predictions with high certainty. The underlying intuition is that a good decision boundary should avoid crossing high-density regions in the target structure [74, 75]. Lower Entropy scores indicate better model performance for this metric.

$$\text{Entropy} = -\frac{1}{n_t} \sum_{i=1}^{n_t} \sum_{j=1}^{C} P_{ij} \log P_{ij}$$

**Information maximization.**  The Entropy score only considers sample-wise certainty, which can be misleading when high-certainty predictions are biased towards a small fraction of classes [19]. To address this challenge, [18] utilize input-output mutual information maximization (InfoMax) [39] as a validation metric. In contrast to Entropy, InfoMax includes an additional class-balance regularization by encouraging the averaged prediction $\bar{p} = \frac{1}{n_t}\sum_{i=1}^{n_t} P_{ij}, \quad \bar{p} \in \mathbb{R}^C$ to be even. Higher InfoMax

scores indicate better model performance according to this metric.

$$\text{InfoMax} = -\sum_{j=1}^{C} \bar{p} \log \bar{p} + \frac{1}{n_t} \sum_{i=1}^{n_t} \sum_{j=1}^{C} P_{ij} \log P_{ij}$$

**Neighborhood consistency.** [19] introduce Soft Neighborhood Density (SND), a novel metric that focuses on the property of neighborhood consistency. SND leverages softmax predictions as features and constructs a sample-to-sample similarity matrix. This matrix is transformed into a probabilistic distribution using the softmax function: $S = \text{softmax}(PP^T/\tau), \quad S \in \mathbb{R}^{n_t \times n_t}$. Here, $\tau$ is a small temperature parameter that sharpens the distribution, enabling the difference between nearby and distant samples. SND favors high neighborhood consistency by prioritizing samples whose predictions are similar to other samples within the same neighborhood, resulting in higher SND scores.

$$\text{SND} = -\frac{1}{n_t} \sum_{i=1}^{n_t} \sum_{j=1}^{n_t} S_{ij} \log S_{ij}$$

**Class correlation.** [20] introduce Corr-C, a class correlation-based metric that evaluates both class diversity and prediction certainty. Corr-C calculates the cosine similarity between the class correlation matrix and an identity matrix. Lower Corr-C scores are indicative of better model performance based on this metric.

$$\text{Corr-C} = \frac{\text{sum}(\text{diag}(P^T P))}{\|P^T P\|_F}$$

We can generally classify model selection baselines into two categories: source domain-based methods, including SourceRisk, IWCV, DEV, and RV, and target domain-specific methods, encompassing Entropy, InfoMax, SND, and Corr-C. Recent model selection studies [19, 18, 20] predominantly align with the target domain-specific approach. This trend arises because access to source data restricts UDA to closed-set UDA and often involves additional model training, making the validation process even more complex than UDA model training. In contrast, target domain-specific methods are more straightforward and effective [19]. EnsV, our proposed method, also falls within the category of target domain-specific methods, but fortunately with enhanced reliability due to a theoretical guarantee designed to avert worst-case model selection scenarios.

## C    Hyperparameter Configurations

In our main experiments, we adopt the setting of previous studies [15, 19] by tuning a single hyperparameter for various UDA methods. The comprehensive hyperparameter settings can be found in Table 13.

## D    Full Model Selection Results

For a comprehensive study, we further consider the parameter weight-based ensemble [53] as our role model, and the EnsV variant based on this role model is denoted as 'EnsV-W'. While the parameter weight-based ensemble also shows competitiveness, it requires all candidate models to share the same architecture and lacks a theoretical guarantee of the ensemble performance. Thus, we recommend the simple and generic prediction-based ensemble, i.e., the default 'EnsV'.

In our experiments, we perform hyperparameter selection for both classification and segmentation tasks. For open-partial-set UDA experiments, we utilize the H-score (%) [69, 70] metric, which combines the accuracy of known classes and unknown samples. For semantic segmentation tasks, we employ the mean intersection-over-union (mIoU) (%) [25, 26] metric. As for other classification tasks, we adopt the accuracy (%) metric. Kindly refer to Table 14 to Table 29 for the complete model selection results.

Table 13: Hyperparameter settings for all considered UDA methods. The settings are partially based on [19], with an expanded search space size from 5 to 7 and the inclusion of additional UDA methods across diverse UDA scenarios.

| UDA method | UDA type | Hyperparameter | Search space | Default value |
|---|---|---|---|---|
| ATDOC [35] | CDA self-training | loss coefficient $\lambda$ | {0.02, 0.05, 0.1, 0.2, 0.5, 1.0, 2.0} | 0.2 |
| BNM [8] | CDA output regularization | loss coefficient $\lambda$ | {0.02, 0.05, 0.1, 0.2, 0.5, 1.0, 2.0} | 1.0 |
| CDAN [6] | CDA feature alignment | loss coefficient $\lambda$ | {0.05, 0.1, 0.2, 0.5, 1.0, 2.0, 5.0} | 1.0 |
| MCC [36] | CDA output regularization | temperature $T$ | {1.0, 1.5, 2.0, 2.5, 3.0, 3.5, 4.0} | 2.5 |
| MDD [33] | CDA output alignment | margin factor $\gamma$ | {0.5, 1.0, 2.0, 3.0, 4.0, 5.0, 6.0} | 4.0 |
| SAFN [7] | CDA/PDA feature regularization | loss coefficient $\lambda$ | {0.002, 0.005, 0.01, 0.02, 0.05, 0.1, 0.2} | 0.05 |
| PADA [10] | PDA feature alignment | loss coefficient $\lambda$ | {0.05, 0.1, 0.2, 0.5, 1.0, 2.0, 5.0} | 1.0 |
| DANCE [11] | OPDA self-supervision | loss coefficient $\eta$ | {0.02, 0.05, 0.1, 0.2, 0.5, 1.0, 2.0} | 0.05 |
| SHOT [12] | white-box SFUDA hypothesis transfer | loss coefficient $\beta$ | {0.03, 0.05, 0.1, 0.3, 0.5, 1.0, 3.0} | 0.3 |
| DINE [35] | black-box SFUDA knowledge distillation | loss coefficient $\beta$ | {0.05, 0.1, 0.2, 0.5, 1.0, 2.0, 5.0} | 1.0 |
| AdaptSeg [25] | segmentation output alignment | loss coefficient $\lambda$ | {0.0001, 0.0003, 0.001, 0.003, 0.01, 0.03} | 0.0002 |
| AdvEnt [26] | segmentation output alignment | loss coefficient $\lambda$ | {0.0001, 0.0003, 0.001, 0.003, 0.01, 0.03} | 0.001 |

Table 14: Validation accuracy (%) of a closed-set UDA method ATDOC [35] on *Office-Home*.

| Method | Ar → Cl | Ar → Pr | Ar → Re | Cl → Ar | Cl → Pr | Cl → Re | Pr → Ar | Pr → Cl | Pr → Re | Re → Ar | Re → Cl | Re → Pr | avg |
|---|---|---|---|---|---|---|---|---|---|---|---|---|---|
| SourceRisk [9] | 51.41 | **77.31** | 78.17 | **66.87** | 74.36 | 75.60 | 61.85 | 48.04 | 76.06 | 71.16 | 58.14 | **84.05** | 68.59 |
| IWCV [14] | 55.88 | 76.57 | 78.88 | 66.25 | 74.50 | 78.33 | 65.60 | 48.04 | 80.58 | **72.06** | 58.14 | 83.87 | 69.89 |
| DEV [15] | 51.41 | 76.55 | 78.88 | 66.25 | 74.36 | 77.67 | 64.77 | 51.29 | 81.62 | 71.16 | **59.98** | 82.43 | 69.70 |
| RV [16] | 56.38 | 76.12 | 80.01 | 66.25 | 76.80 | 78.33 | 67.82 | 55.62 | 80.58 | 71.98 | 56.40 | 83.87 | 70.85 |
| Entropy [17] | 55.88 | 74.14 | 78.88 | 59.25 | 74.52 | 77.67 | 64.19 | 54.39 | 78.54 | 67.57 | 57.23 | 80.96 | 68.60 |
| InfoMax [18] | 55.88 | 74.14 | 78.88 | 59.25 | 77.74 | 77.67 | 64.19 | 54.39 | 78.54 | 67.57 | 56.61 | 80.96 | 68.82 |
| SND [19] | 55.88 | 74.14 | 78.88 | 59.25 | 74.52 | 75.21 | 64.19 | 54.39 | 78.54 | 67.57 | 56.61 | 80.96 | 68.34 |
| Corr-C [20] | 51.41 | 72.00 | 76.04 | 59.37 | 69.36 | 69.54 | 61.85 | 48.04 | 76.06 | 69.30 | 51.71 | 80.31 | 65.42 |
| EnsV-W | **57.85** | 76.57 | **81.04** | 66.25 | 79.48 | 78.52 | 67.94 | 55.62 | 82.17 | 71.9 | 59.24 | 84.03 | 71.72 |
| EnsV | **57.85** | 76.57 | 80.54 | 66.25 | 78.82 | 78.52 | 67.94 | 57.07 | 82.17 | 71.9 | 59.24 | 84.03 | **71.74** |
| Worst | 51.41 | 72.00 | 76.04 | 59.25 | 69.36 | 69.54 | 61.85 | 48.04 | 76.06 | 67.57 | 51.71 | 80.31 | 65.26 |
| Best | 58.01 | 77.31 | 81.04 | 66.91 | 79.48 | 78.52 | 67.94 | 57.07 | 82.17 | 72.06 | 59.98 | 84.03 | 72.04 |

Table 15: Validation accuracy (%) of a closed-set UDA method BNM [8] on *Office-Home*.

| Method | Ar → Cl | Ar → Pr | Ar → Re | Cl → Ar | Cl → Pr | Cl → Re | Pr → Ar | Pr → Cl | Pr → Re | Re → Ar | Re → Cl | Re → Pr | avg |
|---|---|---|---|---|---|---|---|---|---|---|---|---|---|
| SourceRisk [9] | 56.93 | **77.00** | 77.74 | 57.64 | **73.33** | 69.65 | 56.45 | 42.38 | 77.19 | 73.22 | 52.90 | 82.26 | 66.37 |
| IWCV [14] | 46.46 | **77.00** | 79.30 | 63.86 | 61.34 | 62.54 | 63.95 | 42.38 | 78.01 | 71.86 | 55.65 | 83.92 | 65.52 |
| DEV [15] | 57.75 | 71.62 | 79.30 | 57.64 | 67.90 | 75.46 | **66.21** | 54.04 | 78.01 | 73.42 | 57.37 | 82.25 | 68.41 |
| RV [16] | **58.67** | **77.00** | 79.30 | 65.68 | **73.33** | 75.46 | 65.64 | 52.05 | 81.25 | 73.42 | 59.54 | 83.92 | 70.44 |
| Entropy [17] | 53.40 | 67.04 | 78.04 | 63.41 | 71.44 | 73.93 | 63.58 | 52.69 | 80.95 | 71.86 | 57.37 | 83.96 | 68.14 |
| InfoMax [18] | 53.40 | 67.04 | 78.04 | 63.41 | 71.44 | 73.93 | 63.58 | 52.69 | 80.95 | 71.86 | 57.37 | **83.96** | 68.14 |
| SND [19] | 53.40 | 67.04 | 78.04 | 63.41 | 71.44 | 73.93 | 63.58 | 52.69 | 80.95 | 71.86 | 57.37 | **83.96** | 68.14 |
| Corr-C [20] | 46.46 | 67.04 | 74.82 | 49.73 | 61.34 | 62.54 | 56.45 | 42.38 | 74.41 | 68.11 | 47.26 | 78.51 | 60.76 |
| EnsV-W | **58.67** | **77.00** | 80.61 | 66.21 | **73.33** | 76.75 | **66.21** | 53.93 | 81.25 | 73.42 | 57.59 | 83.92 | 70.74 |
| EnsV | **58.67** | **77.00** | 80.61 | 66.21 | **73.33** | 76.75 | **66.21** | 53.93 | 81.25 | 73.42 | 59.54 | 83.92 | **70.90** |
| Worst | 46.46 | 67.04 | 74.82 | 49.73 | 61.34 | 62.54 | 56.45 | 42.38 | 74.41 | 68.11 | 47.26 | 78.51 | 60.75 |
| Best | 58.67 | 77.00 | 80.61 | 67.16 | 74.16 | 76.75 | 66.21 | 54.04 | 81.36 | 73.42 | 59.82 | 84.12 | 71.11 |

Table 16: Validation accuracy (%) of a closed-set UDA method CDAN [6] on *Office-Home*.

| Method | Ar → Cl | Ar → Pr | Ar → Re | Cl → Ar | Cl → Pr | Cl → Re | Pr → Ar | Pr → Cl | Pr → Re | Re → Ar | Re → Cl | Re → Pr | avg |
|---|---|---|---|---|---|---|---|---|---|---|---|---|---|
| SourceRisk [9] | 43.41 | 62.51 | 75.51 | 43.96 | 61.59 | 57.70 | 53.75 | 37.50 | 73.22 | 67.28 | 47.01 | 84.39 | 58.99 |
| IWCV [14] | 43.14 | 62.51 | 77.81 | 44.71 | 54.58 | 56.14 | 65.14 | 37.50 | 81.85 | 74.08 | 43.02 | 84.39 | 60.41 |
| DEV [15] | 57.16 | 71.75 | 77.81 | 62.46 | 55.64 | 71.08 | 65.14 | 56.54 | 81.85 | 74.08 | 57.43 | 78.89 | 67.49 |
| RV [16] | 57.16 | 71.75 | 77.78 | 63.62 | 72.92 | 73.40 | 65.14 | 54.50 | 81.85 | 74.21 | 58.56 | 83.37 | 69.52 |
| Entropy [17] | **57.55** | 72.43 | 77.74 | 63.62 | 72.92 | 73.40 | **65.27** | 56.66 | 81.20 | 74.08 | 58.47 | 83.76 | 69.76 |
| InfoMax [18] | **57.55** | 72.43 | 77.74 | 63.62 | 72.92 | 73.40 | **65.27** | 56.66 | 81.20 | 74.08 | 58.47 | 83.76 | 69.76 |
| SND [19] | **57.55** | 72.43 | 77.78 | 64.61 | **73.73** | 73.40 | 65.14 | 56.66 | 81.85 | 74.08 | 58.47 | 84.73 | 70.04 |
| Corr-C [20] | 43.14 | 63.05 | 73.61 | 43.96 | 54.58 | 56.12 | 51.75 | 37.50 | 73.22 | 65.80 | 43.00 | 77.25 | 56.91 |
| EnsV-W | 57.18 | 73.30 | 77.78 | 63.37 | 73.89 | 73.38 | 65.14 | 55.44 | 81.36 | 73.88 | 58.56 | 84.39 | 69.81 |
| EnsV | **57.55** | **73.71** | 78.33 | 64.61 | **73.73** | 74.39 | 65.14 | 56.56 | 81.85 | 73.88 | 58.56 | 84.73 | 70.25 |
| Worst | 43.14 | 62.51 | 73.61 | 43.96 | 54.58 | 56.12 | 51.63 | 37.50 | 73.22 | 65.80 | 43.00 | 77.25 | 56.86 |
| Best | 57.55 | 73.71 | 78.33 | 64.61 | 73.89 | 74.39 | 65.76 | 56.66 | 81.85 | 74.21 | 59.50 | 84.73 | 70.43 |

Table 17: Validation accuracy (%) of a closed-set UDA method MCC [36] on *Office-Home*.

| Method | Ar → Cl | Ar → Pr | Ar → Re | Cl → Ar | Cl → Pr | Cl → Re | Pr → Ar | Pr → Cl | Pr → Re | Re → Ar | Re → Cl | Re → Pr | avg |
|---|---|---|---|---|---|---|---|---|---|---|---|---|---|
| SourceRisk [9] | 57.23 | 78.19 | **81.75** | 60.65 | 76.50 | **78.79** | 64.15 | 53.15 | 82.17 | **74.91** | 59.20 | 83.96 | 70.89 |
| IWCV [14] | **60.02** | 78.15 | 81.34 | 68.73 | **78.51** | 77.85 | 64.15 | **57.85** | 81.04 | 73.18 | 58.92 | 84.46 | 72.02 |
| DEV [15] | 57.16 | 78.15 | 81.34 | **69.10** | 73.01 | 76.80 | 64.15 | **57.85** | 82.17 | 73.18 | 59.20 | 84.46 | 71.38 |
| RV [16] | 59.34 | **78.53** | 80.70 | **69.10** | 77.83 | 78.22 | **67.20** | **57.85** | 82.24 | **74.91** | 59.20 | **85.54** | 72.56 |
| Entropy [17] | 59.31 | **78.53** | 81.59 | 66.87 | 77.83 | **78.79** | **67.20** | **57.85** | **82.51** | 73.79 | 60.82 | **85.54** | 72.55 |
| InfoMax [18] | **60.02** | 74.66 | **81.75** | 64.98 | 78.24 | 78.49 | 64.15 | 54.52 | 82.19 | 70.62 | 60.89 | 84.46 | 71.25 |
| SND [19] | 53.56 | 77.43 | 79.46 | 67.28 | 76.48 | 76.80 | 65.06 | 54.34 | 81.04 | 74.82 | 58.92 | 85.24 | 70.87 |
| Corr-C [20] | 53.56 | 77.43 | 79.46 | 67.28 | 76.48 | 76.80 | 65.06 | 54.34 | 81.04 | 74.82 | 58.92 | 85.24 | 70.87 |
| EnsV-W | 59.31 | 77.86 | 81.59 | **69.10** | **78.51** | **78.79** | 66.87 | **57.85** | 82.19 | 73.79 | **61.35** | 85.22 | **72.70** |
| EnsV | 59.31 | 77.86 | 81.59 | **69.10** | 77.83 | **78.79** | 66.87 | **57.85** | 82.19 | 73.79 | **61.35** | 85.22 | 72.65 |
| Worst | 53.56 | 73.44 | 79.25 | 60.65 | 73.01 | 75.76 | 59.74 | 53.15 | 79.55 | 67.78 | 57.18 | 82.11 | 67.93 |
| Best | 60.02 | 78.53 | 81.75 | 69.22 | 78.51 | 78.79 | 67.90 | 58.49 | 82.51 | 74.91 | 61.35 | 85.74 | 73.14 |

Table 18: Validation accuracy (%) of a closed-set UDA method MDD [33] on *Office-Home*.

| Method | Ar → Cl | Ar → Pr | Ar → Re | Cl → Ar | Cl → Pr | Cl → Re | Pr → Ar | Pr → Cl | Pr → Re | Re → Ar | Re → Cl | Re → Pr | avg |
|---|---|---|---|---|---|---|---|---|---|---|---|---|---|
| SourceRisk [9] | 54.85 | 73.35 | 77.05 | 58.76 | 69.95 | 72.23 | 60.03 | 51.02 | 77.36 | 68.81 | 57.42 | 82.50 | 66.94 |
| IWCV [14] | 56.40 | 69.52 | 76.59 | 58.76 | 67.40 | 69.43 | 61.89 | 56.43 | 76.82 | 71.94 | 56.68 | **84.43** | 67.19 |
| DEV [15] | 57.71 | **75.42** | 77.05 | 58.76 | **72.99** | 70.51 | **63.95** | 56.43 | 80.26 | 70.54 | 56.68 | 82.14 | 68.54 |
| RV [16] | **58.05** | **75.42** | 76.59 | 63.54 | 69.95 | **73.74** | **63.95** | 51.02 | **80.38** | 72.23 | 58.17 | **84.43** | 68.96 |
| Entropy [17] | 57.73 | 74.54 | **78.22** | **64.07** | **72.99** | **73.74** | **63.95** | 55.85 | **80.38** | 71.61 | 59.31 | 84.28 | 69.72 |
| InfoMax [18] | **58.05** | 74.54 | **78.22** | **64.07** | **72.99** | **73.74** | **63.95** | 55.85 | **80.38** | 71.61 | 59.31 | 84.28 | **69.75** |
| SND [19] | **58.05** | **75.42** | 77.05 | 44.99 | **72.99** | 48.06 | 37.08 | 21.60 | 80.26 | 71.94 | 34.39 | **84.43** | 58.86 |
| Corr-C [20] | 39.08 | 59.74 | 69.61 | 44.99 | 54.58 | 48.06 | 37.08 | 21.60 | 64.22 | 61.31 | 34.39 | 75.87 | 50.88 |
| EnsV-W | 54.89 | **75.42** | 78.01 | 61.89 | **72.99** | 72.23 | 63.08 | 56.43 | 79.66 | **72.23** | 60.02 | 83.96 | 69.23 |
| EnsV | 56.40 | **75.42** | 77.05 | **64.07** | **72.99** | 72.23 | 63.08 | **57.02** | 80.26 | **72.23** | 60.02 | **84.43** | 69.60 |
| Worst | 39.08 | 59.74 | 69.61 | 44.99 | 54.58 | 48.06 | 37.08 | 21.60 | 64.22 | 61.31 | 34.39 | 75.87 | 50.88 |
| Best | 58.05 | 75.42 | 78.22 | 64.07 | 72.99 | 73.74 | 63.95 | 57.02 | 80.38 | 72.23 | 60.02 | 84.43 | 70.04 |

Table 19: Validation accuracy (%) of a closed-set UDA method SAFN [7] on *Office-Home*.

| Method | Ar → Cl | Ar → Pr | Ar → Re | Cl → Ar | Cl → Pr | Cl → Re | Pr → Ar | Pr → Cl | Pr → Re | Re → Ar | Re → Cl | Re → Pr | avg |
|---|---|---|---|---|---|---|---|---|---|---|---|---|---|
| SourceRisk [9] | 50.78 | 69.72 | 76.06 | 59.66 | 70.29 | 69.86 | 60.90 | 46.07 | **77.71** | 70.05 | **57.16** | 80.96 | 65.77 |
| IWCV [14] | 50.24 | 69.72 | 77.28 | 62.63 | 67.24 | 69.86 | 58.84 | 49.69 | 75.72 | 71.45 | **57.16** | 79.97 | 65.82 |
| DEV [15] | 51.07 | 69.72 | 76.64 | 59.66 | 71.26 | 70.44 | 58.84 | 49.69 | 75.72 | 70.95 | 50.65 | 76.64 | 64.84 |
| RV [16] | 51.07 | 71.41 | 76.64 | 62.63 | 68.44 | 70.44 | 58.84 | 44.49 | **77.71** | 71.45 | 54.82 | **81.46** | 65.78 |
| Entropy [17] | 45.93 | 69.72 | 75.49 | 55.29 | 67.22 | 68.35 | 54.26 | 43.30 | 75.69 | 70.00 | 49.99 | 80.60 | 62.99 |
| InfoMax [18] | 50.47 | 64.52 | 75.49 | 62.46 | 68.35 | **70.98** | 61.23 | 43.30 | 75.69 | 70.00 | 55.37 | 80.60 | 65.31 |
| SND [19] | 45.93 | 64.36 | 70.60 | 55.29 | 60.13 | 62.50 | 54.26 | 43.30 | 71.43 | 64.15 | 49.99 | 76.64 | 59.88 |
| Corr-C [20] | 45.93 | 69.72 | 76.60 | 55.29 | 60.13 | 62.50 | 61.23 | 43.30 | 71.43 | **71.45** | 49.99 | 76.64 | 61.52 |
| EnsV-W | **51.73** | 72.07 | 76.64 | **64.65** | **70.98** | 71.26 | **63.66** | **50.52** | 77.48 | 70.99 | **57.16** | **81.46** | **67.38** |
| EnsV | 51.07 | **72.27** | **77.30** | 63.58 | 70.29 | **71.70** | 62.71 | 49.69 | **77.71** | 71.45 | 55.78 | 80.96 | 67.04 |
| Worst | 45.93 | 64.36 | 70.60 | 55.29 | 60.13 | 62.50 | 54.26 | 43.30 | 71.43 | 64.15 | 49.99 | 76.64 | 59.88 |
| Best | 51.73 | 72.27 | 77.30 | 64.65 | 70.98 | 71.70 | 63.66 | 50.52 | 77.71 | 71.45 | 57.16 | 81.46 | 67.55 |

Table 20: Validation accuracy (%) of closed-set UDA methods on *Office-31*.

| Method | ATDOC [35] | | | | | BNM [8] | | | | | CDAN [6] | | | | |
|---|---|---|---|---|---|---|---|---|---|---|---|---|---|---|---|
| | A → D | A → W | D → A | W → A | avg | A → D | A → W | D → A | W → A | avg | A → D | A → W | D → A | W → A | avg |
| SourceRisk [9] | 88.96 | **87.80** | 73.65 | 71.46 | 80.47 | **90.36** | **89.43** | 73.13 | 72.70 | 81.41 | 91.16 | **89.06** | 66.33 | 61.46 | 77.00 |
| IWCV [14] | 86.14 | 86.54 | 73.65 | 71.46 | 79.45 | 85.54 | **89.43** | 73.13 | 72.70 | 80.20 | 69.08 | 58.74 | 66.33 | 61.46 | 63.90 |
| DEV [15] | 86.14 | 86.54 | 73.65 | 71.46 | 79.45 | 85.54 | **89.43** | 73.13 | 72.70 | 80.20 | 91.16 | 88.30 | 66.33 | 61.46 | 76.81 |
| RV [16] | 89.96 | 87.23 | 74.28 | **75.58** | 81.76 | 88.55 | **89.43** | 74.90 | 66.52 | 79.85 | 91.16 | 88.30 | **76.18** | **70.36** | 81.50 |
| Entropy [17] | 86.14 | **87.80** | 73.87 | 72.70 | 80.13 | 85.54 | 83.14 | 71.07 | 74.26 | 78.50 | 91.16 | **89.06** | 72.88 | **70.36** | 80.87 |
| InfoMax [18] | 86.14 | **87.80** | 73.87 | 72.70 | 80.13 | 85.54 | 83.14 | 71.07 | 69.97 | 77.43 | 91.16 | 88.30 | 72.88 | **70.36** | 80.68 |
| SND [19] | **92.37** | **87.80** | 73.87 | 72.70 | 81.69 | 85.54 | 83.14 | 74.62 | 74.26 | 79.39 | 92.37 | 88.55 | 72.88 | 70.22 | 81.01 |
| Corr-C [20] | 90.96 | 84.40 | 71.88 | 70.22 | 79.37 | 84.34 | 78.99 | 67.80 | 66.52 | 74.41 | 67.67 | 59.62 | 58.15 | 58.43 | 60.97 |
| EnsV-W | **92.37** | **87.80** | **74.65** | 75.01 | **82.46** | 88.55 | **89.43** | **75.43** | **75.29** | 82.18 | **92.77** | 88.55 | **76.18** | 70.22 | **81.93** |
| EnsV | 90.96 | **87.80** | **74.65** | 75.01 | 82.11 | **90.36** | **89.43** | **75.43** | 74.30 | **82.38** | **92.77** | 88.55 | **76.18** | 70.22 | **81.93** |
| Worst | 86.14 | 84.40 | 71.88 | 70.22 | 78.16 | 84.34 | 78.99 | 67.80 | 66.52 | 74.41 | 67.67 | 57.11 | 58.15 | 58.43 | 60.34 |
| Best | 92.37 | 87.80 | 75.04 | 75.58 | 82.70 | 90.36 | 89.43 | 75.75 | 75.29 | 82.71 | 92.77 | 89.06 | 76.18 | 70.57 | 82.15 |

Table 21: Validation accuracy (%) of closed-set UDA methods on *Office-31*.

| Method | MCC [36] | | | | | MDD [33] | | | | | SAFN [7] | | | | |
|---|---|---|---|---|---|---|---|---|---|---|---|---|---|---|---|
| | A → D | A → W | D → A | W → A | avg | A → D | A → W | D → A | W → A | avg | A → D | A → W | D → A | W → A | avg |
| SourceRisk [9] | 90.96 | 91.07 | 73.33 | 72.89 | 82.06 | 91.06 | 86.23 | 76.68 | 74.76 | 82.18 | 83.73 | 87.17 | 68.96 | 69.44 | 77.33 |
| IWCV [14] | 91.16 | 88.55 | 73.33 | 72.89 | 81.48 | 91.16 | 89.18 | 76.68 | 74.30 | 82.83 | 86.55 | 80.38 | 68.96 | **69.68** | 76.39 |
| DEV [15] | 89.16 | 93.08 | 73.33 | 72.06 | 81.91 | 91.16 | 89.18 | 76.68 | 74.62 | 82.91 | 86.55 | 80.38 | 68.96 | 67.45 | 75.84 |
| RV [16] | 89.06 | 93.08 | 74.42 | 73.52 | 82.52 | **92.57** | 86.79 | 73.91 | **74.97** | 82.07 | 90.83 | 87.17 | 68.76 | 68.62 | 78.85 |
| Entropy [17] | 90.56 | **93.46** | **74.83** | 73.02 | 82.97 | **92.57** | **90.82** | 78.03 | 74.58 | 84.00 | **91.57** | 85.66 | 67.20 | 69.26 | 78.42 |
| InfoMax [18] | 89.16 | 88.55 | 74.16 | **73.70** | 81.39 | **92.57** | **90.82** | 78.03 | **74.97** | **84.10** | **91.57** | **87.42** | 67.20 | 69.26 | 78.86 |
| SND [19] | **91.97** | **93.46** | **74.83** | 73.02 | **83.32** | 92.17 | **90.82** | 78.03 | **74.97** | 84.00 | 89.96 | 85.66 | 67.20 | 69.26 | 78.02 |
| Corr-C [20] | 91.37 | **93.46** | **74.83** | 73.02 | 83.17 | 91.57 | 85.66 | 73.91 | 74.58 | 81.43 | 86.75 | 80.38 | 67.09 | **69.68** | 75.98 |
| EnsV-W | 90.56 | 91.07 | 74.16 | **73.70** | 82.37 | **92.57** | **90.82** | 77.53 | 74.30 | 83.80 | **91.57** | 87.17 | **70.22** | 69.12 | **79.52** |
| EnsV | 90.56 | 91.45 | 73.80 | **73.70** | 82.38 | **92.57** | **90.82** | 77.53 | 74.30 | 83.80 | 90.96 | 87.17 | **70.22** | 69.12 | 79.37 |
| Worst | 86.75 | 87.17 | 71.18 | 69.93 | 78.76 | 87.35 | 85.66 | 73.91 | 72.20 | 79.78 | 83.73 | 80.38 | 67.09 | 67.45 | 74.66 |
| Best | 91.97 | 93.46 | 74.83 | 74.01 | 83.57 | 92.57 | 92.20 | 78.03 | 75.01 | 84.45 | 91.57 | 87.42 | 70.43 | 69.68 | 79.78 |

Table 22: Validation accuracy (%) of a closed-set UDA method CDAN [6] on *DomainNet-126*.

| Method | C → S | P → C | P → R | R → C | R → P | R → S | S → P | avg |
|---|---|---|---|---|---|---|---|---|
| Entropy [17] | **58.04** | **64.78** | 74.42 | **69.39** | **68.65** | **60.63** | **62.94** | **65.55** |
| InfoMax [18] | **58.04** | **64.78** | 74.42 | **69.39** | **68.65** | **60.63** | **62.94** | **65.55** |
| SND [19] | **58.04** | **64.78** | 74.42 | **69.39** | **68.65** | **60.63** | 60.70 | 65.23 |
| Corr-C [20] | **58.04** | 57.73 | 74.42 | 56.98 | 65.07 | 51.23 | 60.70 | 60.60 |
| EnsV-W | 55.15 | 60.98 | 73.86 | 60.99 | 65.07 | 55.50 | 60.27 | 61.69 |
| EnsV | 56.73 | 64.67 | **74.44** | 67.08 | 67.97 | 58.12 | 62.57 | 64.51 |
| Worst | 51.59 | 57.73 | 73.44 | 56.98 | 63.06 | 51.23 | 58.46 | 58.93 |
| Best | 58.04 | 64.78 | 74.44 | 69.39 | 68.65 | 60.63 | 62.94 | 65.55 |

Table 23: Validation accuracy (%) of a closed-set UDA method BNM [8] on *DomainNet-126*.

| Method | C → S | P → C | P → R | R → C | R → P | R → S | S → P | avg |
|---|---|---|---|---|---|---|---|---|
| Entropy [17] | 56.42 | 61.57 | 74.31 | 65.15 | 65.15 | 40.95 | 63.42 | 61.00 |
| InfoMax [18] | 56.42 | 68.95 | 74.31 | 65.15 | 65.15 | 54.93 | 63.42 | 64.05 |
| SND [19] | 43.78 | 61.57 | 74.31 | 51.55 | 54.40 | 40.95 | 54.59 | 54.45 |
| Corr-C [20] | 43.78 | 60.03 | **77.62** | 59.47 | 67.19 | 40.95 | 59.64 | 58.38 |
| EnsV-W | **58.48** | 68.42 | **77.62** | 66.05 | 67.79 | 57.65 | 64.34 | 65.76 |
| EnsV | 57.73 | **69.63** | **77.62** | **66.10** | 67.79 | 57.65 | 64.34 | **65.84** |
| Worst | 43.78 | 60.03 | 74.31 | 51.55 | 54.40 | 40.95 | 54.59 | 54.23 |
| Best | 58.48 | 69.63 | 78.68 | 66.10 | 67.79 | 58.50 | 65.20 | 66.34 |

Table 24: Validation accuracy (%) of a closed-set UDA method ATDOC [35] on *DomainNet-126*.

| Method | C → S | P → C | P → R | R → C | R → P | R → S | S → P | avg |
|---|---|---|---|---|---|---|---|---|
| Entropy [17] | 46.43 | 65.98 | 79.60 | 61.52 | 64.24 | 57.92 | 59.46 | 62.16 |
| InfoMax [18] | 46.43 | 65.98 | 79.60 | 61.52 | 64.24 | 57.92 | 59.46 | 62.16 |
| SND [19] | 46.43 | 65.98 | 79.60 | 61.52 | 64.24 | 47.58 | 59.46 | 60.69 |
| Corr-C [20] | 54.71 | 60.63 | 74.42 | 59.33 | 64.58 | 52.66 | 59.95 | 60.90 |
| EnsV-W | **63.12** | 69.57 | 78.33 | 67.93 | 69.32 | 60.85 | 66.33 | 67.92 |
| EnsV | 62.11 | **71.14** | **80.01** | **69.45** | **69.79** | **61.35** | **67.10** | **68.71** |
| Worst | 46.43 | 60.63 | 74.42 | 59.33 | 64.24 | 47.58 | 59.46 | 58.87 |
| Best | 63.12 | 71.14 | 80.38 | 69.45 | 69.79 | 61.35 | 67.10 | 68.90 |

Table 25: Validation accuracy (%) of a partial-set UDA method PADA [10] on *Office-Home*.

| Method | Ar → Cl | Ar → Pr | Ar → Re | Cl → Ar | Cl → Pr | Cl → Re | Pr → Ar | Pr → Cl | Pr → Re | Re → Ar | Re → Cl | Re → Pr | avg |
|---|---|---|---|---|---|---|---|---|---|---|---|---|---|
| SourceRisk [9] | 45.03 | 68.85 | 81.89 | 43.25 | 46.83 | 57.26 | 57.21 | 36.42 | 76.53 | 71.26 | 44.30 | 77.76 | 58.87 |
| IWCV [14] | **55.58** | 65.10 | 84.54 | 51.42 | **61.29** | 53.01 | 56.93 | 35.16 | 81.34 | 70.52 | **60.78** | 74.12 | 62.49 |
| DEV [15] | 54.81 | **78.15** | 78.02 | **58.13** | 61.29 | 50.14 | 67.86 | 35.16 | 83.21 | 74.66 | 57.91 | 77.76 | 64.76 |
| RV [16] | 43.22 | 65.10 | 81.89 | 42.70 | 48.74 | 52.79 | 57.21 | 35.16 | 77.80 | 73.46 | 44.30 | 77.76 | 58.34 |
| Entropy [17] | 40.12 | 40.11 | 55.94 | 52.43 | 37.25 | 50.14 | 57.30 | 47.22 | 81.34 | 70.52 | 52.18 | 82.13 | 55.56 |
| InfoMax [18] | 54.81 | 69.24 | 78.02 | 52.43 | 37.25 | 57.30 | 47.22 | 71.84 | 70.52 | 52.18 | 74.12 | 59.59 |
| SND [19] | 40.12 | 40.11 | 55.94 | 58.13 | 56.13 | 64.11 | 70.62 | **51.22** | 81.34 | 74.66 | **60.78** | 82.13 | 61.27 |
| Corr-C [20] | 40.12 | 40.11 | 55.94 | 54.18 | 46.89 | 53.01 | 58.59 | 38.93 | 77.80 | 71.26 | 57.91 | 77.70 | 56.04 |
| EnsV-W | **55.58** | 77.25 | 86.14 | 58.13 | 60.17 | 67.86 | 73.00 | 37.97 | 84.04 | 76.77 | 57.91 | 83.75 | **68.21** |
| EnsV | 54.81 | 69.24 | **86.53** | **58.13** | 56.13 | 64.11 | 70.62 | 51.22 | **84.04** | 76.86 | 60.78 | **84.20** | 68.06 |
| Worst | 40.12 | 40.11 | 55.94 | 41.41 | 37.25 | 50.14 | 56.93 | 34.87 | 71.84 | 70.52 | 44.24 | 74.12 | 51.46 |
| Best | 55.58 | 78.15 | 86.53 | 58.13 | 61.29 | 68.19 | 73.00 | 51.22 | 84.04 | 76.86 | 60.78 | 84.20 | 69.83 |

Table 26: Validation accuracy (%) of a partial-set UDA method SAFN [7] on *Office-Home*.

| Method | Ar → Cl | Ar → Pr | Ar → Re | Cl → Ar | Cl → Pr | Cl → Re | Pr → Ar | Pr → Cl | Pr → Re | Re → Ar | Re → Cl | Re → Pr | avg |
|---|---|---|---|---|---|---|---|---|---|---|---|---|---|
| SourceRisk [9] | **59.40** | 77.14 | 81.34 | 63.97 | 67.00 | 71.29 | 65.60 | 46.21 | 76.81 | 70.89 | 58.51 | 79.10 | 68.11 |
| IWCV [14] | 52.24 | 74.45 | **82.16** | 70.98 | 62.41 | 70.18 | 63.45 | 53.49 | 76.81 | 73.65 | 56.00 | 78.49 | 67.86 |
| DEV [15] | 55.22 | 74.45 | 80.07 | 70.98 | 67.00 | 71.29 | 63.45 | 51.70 | 76.81 | 73.65 | 57.91 | 80.39 | 68.58 |
| RV [16] | 53.67 | 71.60 | 81.34 | 67.58 | 67.00 | 73.27 | 65.70 | 48.54 | 76.81 | 73.65 | 56.00 | 79.89 | 67.92 |
| Entropy [17] | 58.93 | 74.90 | 80.73 | 70.98 | **74.12** | 69.80 | 70.16 | 50.09 | 79.24 | 74.10 | 57.85 | 80.06 | 70.08 |
| InfoMax [18] | 51.82 | 67.62 | 76.97 | 64.65 | 65.77 | 69.80 | 59.69 | 50.09 | 74.10 | 66.67 | 53.31 | 75.52 | 64.67 |
| SND [19] | 51.82 | 74.90 | 80.73 | 70.98 | **74.12** | **75.10** | 70.16 | 50.09 | 79.24 | 74.10 | 53.31 | 80.06 | 69.55 |
| Corr-C [20] | **59.40** | **77.20** | **82.16** | 67.58 | 72.89 | **75.10** | 70.16 | **55.70** | 80.12 | **75.94** | 52.00 | **80.73** | 70.75 |
| EnsV-W | **59.40** | **77.20** | **82.16** | **71.72** | 72.89 | 74.82 | **72.45** | **55.70** | **80.73** | **75.94** | **59.16** | **80.73** | **71.91** |
| EnsV | 55.22 | 76.30 | 81.28 | 67.58 | 72.89 | 74.05 | 54.63 | 80.12 | 74.05 | 58.51 | 80.39 | 70.31 |
| Worst | 51.52 | 67.62 | 76.97 | 61.07 | 62.35 | 69.80 | 59.69 | 46.21 | 74.10 | 66.67 | 52.00 | 75.52 | 63.63 |
| Best | 59.40 | 77.20 | 82.16 | 71.72 | 74.12 | 75.10 | 72.45 | 55.70 | 80.73 | 75.94 | 59.16 | 80.73 | 72.03 |

Table 27: H-score [69, 70] (%) of an open-partial-set UDA method DANCE [11] on *Office-Home*.

| Method | Ar → Cl | Ar → Pr | Ar → Re | Cl → Ar | Cl → Pr | Cl → Re | Pr → Ar | Pr → Cl | Pr → Re | Re → Ar | Re → Cl | Re → Pr | avg |
|---|---|---|---|---|---|---|---|---|---|---|---|---|---|
| Entropy [17] | 38.29 | 26.08 | 36.51 | 32.92 | 17.10 | 32.19 | 37.69 | 46.40 | 45.53 | 25.39 | 33.75 | 39.37 | 34.27 |
| InfoMax [18] | 38.29 | 26.08 | 36.51 | 32.92 | 17.10 | 32.19 | 37.69 | 46.40 | 45.33 | 25.39 | 33.75 | 39.37 | 34.25 |
| SND [19] | 1.00 | 0.00 | 12.73 | 0.00 | 42.84 | 1.95 | 19.77 | 11.99 | 35.69 | 25.39 | 0.00 | 28.40 | 14.98 |
| Corr-C [20] | 1.00 | 0.00 | 12.73 | 0.00 | 42.84 | 1.95 | 19.77 | 11.99 | 35.69 | 69.02 | 0.00 | 28.40 | 18.62 |
| EnsV-W | **67.00** | 75.15 | 66.57 | 67.87 | 67.35 | 59.05 | 66.41 | **62.59** | **69.40** | 59.86 | 67.54 | 73.40 | 66.85 |
| EnsV | 38.40 | **76.96** | 66.57 | **71.76** | **75.17** | **69.99** | **77.42** | 48.15 | **69.40** | **81.84** | **67.54** | **84.31** | **68.96** |
| Worst | 1.00 | 0.00 | 12.73 | 0.00 | 17.10 | 1.95 | 19.77 | 11.99 | 35.69 | 25.39 | 0.00 | 28.40 | 12.84 |
| Best | 67.00 | 76.96 | 66.57 | 71.76 | 75.17 | 69.99 | 77.42 | 64.32 | 72.87 | 81.84 | 67.54 | 84.31 | 72.98 |

Table 28: Validation accuracy (%) of a white-box source-free UDA method SHOT [12] on *Office-Home*.

| Method | Ar → Cl | Ar → Pr | Ar → Re | Cl → Ar | Cl → Pr | Cl → Re | Pr → Ar | Pr → Cl | Pr → Re | Re → Ar | Re → Cl | Re → Pr | avg |
|---|---|---|---|---|---|---|---|---|---|---|---|---|---|
| Entropy [17] | 49.14 | 76.17 | 79.23 | 60.57 | 73.94 | 74.00 | 60.69 | 48.66 | 79.73 | 68.89 | 53.56 | 81.93 | 67.21 |
| InfoMax [18] | 49.14 | 76.17 | 79.23 | 60.57 | 73.94 | 74.00 | 60.69 | 48.66 | 79.73 | 68.89 | 53.56 | 81.93 | 67.21 |
| SND [19] | 49.14 | 76.17 | 79.23 | 60.57 | 76.59 | 74.00 | 64.28 | **54.55** | 79.73 | 68.89 | 58.81 | 81.93 | 68.66 |
| Corr-C [20] | 55.60 | 76.66 | 79.83 | 67.04 | 76.59 | 76.86 | 66.63 | **54.55** | 80.74 | **73.71** | 58.81 | **84.61** | 70.97 |
| EnsV-W | **56.36** | **77.81** | **81.36** | **68.27** | **78.78** | **78.91** | 65.80 | 54.52 | **82.01** | 73.01 | **59.45** | **84.61** | 71.74 |
| EnsV | **56.36** | **77.81** | **81.36** | **68.27** | **78.78** | **78.91** | **67.12** | 54.52 | **82.01** | 73.34 | **59.45** | **84.61** | **71.88** |
| Worst | 49.14 | 76.17 | 79.23 | 60.57 | 73.94 | 74.00 | 60.69 | 48.66 | 79.73 | 68.89 | 53.56 | 81.93 | 67.21 |
| Best | 56.36 | 77.95 | 81.36 | 68.27 | 79.05 | 78.91 | 67.33 | 55.33 | 82.01 | 73.88 | 59.54 | 84.66 | 72.05 |

Table 29: Validation accuracy (%) of a white-box source-free UDA method SHOT [12] on *Office-31*.

| Method | A → D | A → W | D → A | W → A | avg |
|---|---|---|---|---|---|
| Entropy [17] | 90.76 | 88.68 | 71.21 | 72.13 | 80.69 |
| InfoMax [18] | 90.76 | 88.68 | 71.21 | 72.13 | 80.69 |
| SND [19] | 90.76 | 88.68 | 71.21 | 72.13 | 80.69 |
| Corr-C [20] | 90.76 | 90.19 | 71.21 | 71.96 | 81.03 |
| EnsV-W | **94.78** | **91.82** | **75.15** | **74.55** | **84.08** |
| EnsV | **94.78** | **91.82** | **75.15** | **74.55** | **84.08** |
| Worst | 90.76 | 88.68 | 71.21 | 71.92 | 80.64 |
| Best | 94.78 | 93.33 | 75.58 | 74.55 | 84.56 |

