# OpenReview forum: "Towards Reliable Model Selection for Unsupervised Domain Adaptation: An Empirical Study and A Certified Baseline"
_NeurIPS.cc/2024/Datasets_and_Benchmarks_Track — NeurIPS 2024 Track Datasets and Benchmarks Poster_

### Official Review · Reviewer_yasW · 2024-07-22
**Good baseline for future research**

**Rating:** 7
**Confidence:** 2
**Correctness:** The paper's claims are supported by e…

**Review:**

### Quality

The paper is generally well-organized and easy to read. In the field of UDA, where many variants of problem settings exist, the paper deals with all major problem settings rather than focusing on a specific problem setting, enhancing the paper's value.

### Clarity

The motivation is clear and the ensemble method proposed as a new baseline is also clear. The clarity of this paper is high.

### Originality

While there have been several papers on model selection for UDA, none have focused on a broad problem set and reevaluation of existing methods, and there is originality in the paper's findings.

### Significance

There are many methods in UDA that are sensitive to hyperparameters, and several that have poor reproducibility. In this context, this paper provides a baseline that can serve as a standard for model evaluation as well as model selection for UDA. Subsequent studies will be able to reasonably evaluate their methods using the evaluation protocols provided by this paper. Overall, the implications for the field of UDA are expected to be significant.

**Strengths:**

- The paper re-evaluates model selection in UDA, which is difficult due to the absence of a labeled target dataset, and reveals the risk that existing methods select suboptimal models.
- The ensemble-based model selection method proposed in the paper is simple and easy to implement.
- The paper conducts experiments in extensive combinations of settings and shows that the proposed baseline outperforms most existing methods, making it a fundamental baseline for future work.

**Additional Feedback:**

There is nothing to mention.

**Clarity:**

The motivation is clear and the ensemble method proposed as a new baseline is also clear. The clarity of this paper is high.

**Documentation:**

There is nothing to mention.

**Ethics:**

There is nothing to mention.

**Limitations:**

Although limitation is mentioned in the checklist, no discussion of limitation can be found in the main text.

**Opportunities For Improvement:**

- Please discuss limitations. The paper responds that the checklist discusses limitations, but I could not find where this is explicitly discussed in the main text.
- The ensemble method proposed by the paper seems to assume the existence of a hyperparameter that achieves good performance. Also, the worst-case model used in the experiments achieves some performance. In practice, especially in the early phases of model development, the search range of hyperparameters may be inappropriate, and none of the hyperparameters achieve good performance. Investigating the practicality of the proposed method in such situations may help enhance the paper's claims.

**Relation To Prior Work:**

The existing studies presented in Section 2 are sufficiently comprehensive and reasonably describe the position of this paper.

**Summary And Contributions:**

The paper re-evaluates existing methods in various settings in the context of model selection for unsupervised domain adaptation (UDA). The main findings of the paper are that (i) most existing model selection methods cannot avoid the risk of selecting the worst-case model and (ii) a simple ensemble-based baseline can avoid worst-case model selection and achieve almost best target performance. The paper's evaluation is extensive across a wide range of experimental settings and combinations of existing methods, including source-free DA, and is expected to serve as a fundamental benchmark for subsequent studies.

---

> ### Author Rebuttal · Authors · 2024-08-16
>
> Thank you very much for your positive rating and for the in-depth review of our paper. We sincerely appreciate your helpful feedback.
>
> > **Q1**: Please discuss limitations. The paper responds that the checklist discusses limitations, but I could not find where this is explicitly discussed in the main text.
>
> **A1**: Thank you for your valuable feedback. Although we have touched upon the limitations in the main text, including the last sentence of the conclusion, we understand the need for a more explicit discussion. As you suggested, **we will include a detailed discussion** of the limitations in the revised version.
>
> EnsV selects the candidate model that most closely matches the ensemble's performance. While EnsV consistently avoids selecting the worst-performing models, its performance can be suboptimal if the ensemble (role model) itself is suboptimal. **EnsV may face challenges** in the following scenarios:
> - **Single High-Performing vs. Poor Model**: When the model pool contains only one high-performing model and one poor model, EnsV may struggle. In such cases, the ensemble of just these two models might not accurately reflect the quality of the good model.
> - **Small Performance Differences**: If the performance differences among all candidate models are small, EnsV may have difficulty distinguishing between them. In such scenarios, using ensemble results for model selection may not provide the fine-grained control needed to accurately differentiate between the candidates.
> - **Predominantly Poor Models**: When the majority of models in the pool are poor, with only a few having normal or good performance, EnsV may encounter issues. An ensemble composed mostly of poor models may produce results that are closer to the poor models, leading to a final selection of a suboptimal candidate.
>
> > **Q2**: The ensemble method proposed by the paper seems to assume the existence of a hyperparameter that achieves good performance. Also, the worst-case model used in the experiments achieves some performance. In practice, especially in the early phases of model development, the search range of hyperparameters may be inappropriate, and none of the hyperparameters achieve good performance. Investigating the practicality of the proposed method in such situations may help enhance the paper's claims.
>
> **A2**: Thank you for the constructive feedback. **We have followed your advice and evaluated our EnsV** method and the state-of-the-art selection method SND in a scenario where the **initial hyperparameter range is inappropriate and many candidate models perform similarly to random trials**.
>
> Typically, domain adaptation methods use a pre-trained backbone, such as an ImageNet-pretrained feature extractor, with a randomly initialized classifier on top. Following CDAN, many studies use a base learning rate (LR) of 0.01 for the newly added classifier and 0.001 for the backbone, which has become a common setting in domain adaptation. We explored classifier **learning rates in the range of {0.01, 0.03, 0.1}**. For each LR setting, we saved five checkpoints every 2000 iterations, resulting in **a pool of 15 candidate models**. Below, we present the actual target accuracy, the EnsV selection score (higher is better), and the SND selection score (higher is better) for these models and mark the best selection according to respective selection metrics/scores.
>
> The following table shows the actual target-domain accuracy (%) of the 15 checkpoints. We find that using the default learning rate achieves normal model performance, whereas larger learning rates such as 0.03 can lead to model collapse, resulting in performance at a random trial level. **The best model has an accuracy of 55.51%.**
> | Iteration-LR   | 0.01 | 0.03   | 0.1 |
> | :---        |    ----:   |     ----:   |     ----:   |
> | 1998   | 52.00 | 3.89 | 2.25 |
> | 3996   | **55.51** | 3.76 | 2.27 |
> | 5994   | 54.34 | 5.04 | 2.29 |
> | 7992   | 53.79 | 6.67 | 2.27 |
> | 9990   | 54.96 | 7.35 | 2.20 |
>
> The following table illustrates the situation when using our EnsV for model selection, with scores calculated according to Figure 1 in our paper. The score represents model accuracy (%) when using the ensembles as the ground truth, and EnsV selects the model with the highest score. EnsV selected the model 9990-0.01 as the final choice. According to the actual accuracy table, **the model selection performance of EnsV is 54.96%.**
> | Iteration-LR   | 0.01 | 0.03   | 0.1 |
> | :---        |    ----:   |     ----:   |     ----:   |
> | 1998   | 69.46 | 11.00 | 5.43 |
> | 3996   | 74.18 | 12.88 | 3.99 |
> | 5994   | 74.43 | 17.48 | 3.64 |
> | 7992   | 74.41 | 21.72 | 5.57 |
> | 9990   | **75.21** | 27.15 | 4.01 |
>
> The following table shows the situation when using the state-of-the-art model selection method SND, with scores calculated following the guidance detailed in our appendix. SND selects the model with the highest score. We find that **SND selects** the model 5994-0.1 as the final choice, which is **a collapsed model with an actual accuracy of 2.29%.**
>
> | Iteration-LR   | 0.01 | 0.03   | 0.1 |
> | :---        |    ----:   |     ----:   |     ----:   |
> | 1998   | 3.18 | 6.21 | 8.16 |
> | 3996   | 2.84 | 5.24 | 8.26 |
> | 5994   | 2.95 | 4.44 | **8.32** |
> | 7992   | 3.05 | 3.99 | 7.89 |
> | 9990   | 3.03 | 3.23 | 7.96 |
>
> Note that this scenario is very challenging for EnsV because **2/3 of the models have random trial accuracy**. We believe that extending the range of learning rate (LR) hyperparameter values to include a broader set such as {0.001, 0.003, 0.01, 0.03, 0.1} would benefit EnsV rather than challenge it. Smaller LRs are likely to lead to suboptimally optimized models with some performance rather than collapsed models, which has been demonstrated in our submission to be advantageous for EnsV.

---

### Official Review · Reviewer_deqf · 2024-07-24
**This paper addresses an important aspect of UDA training and proposes a simple solution with limited analysis**

**Rating:** 6
**Confidence:** 4
**Correctness:** Yes.

**Review:**

This paper addresses an important and oft-overlooked aspect of UDA training, which is how to select the best model (hyperparameter) for your dataset?  While being of great practical significance in the real-world application of UDA algorithms, it is often handwaved away and left undiscussed.  This paper proposes a simple and seemingly effective strategy, use the ensemble target prediction of all candidate models to select the best model for use.  This requires the full training of all m possible models, which seems like a huge cost but is apparently inherent in the problem setting.  On the various UDA classification benchmarks reported in this paper, the ensemble consistently performs significantly better than other model selection methods and is close to the best-case upper bound.  Despite these good results, however, I find that the proposed method too simplistic  and the analysis to be limited.

____________________________________________________________________________________________________
Upon reading the author rebuttal and other reviewer comments, I agree that I overlooked the benchmarking aspect of the paper.  For **this reason and this reason alone**, I change my recommendation from 4 to 6: borderline accept.  My concerns about the baseline method and problem remain in spite of the author rebuttal.

**Strengths:**

This paper addresses the important issue of UDA model selection and proposes an effective solution of uses the candidate model ensemble for model selection.

**Additional Feedback:**

No additional feedback.

**Clarity:**

The paper could use a little more clarification on the worst-case analysis in table 1.  I am assuming the worst case is when the 1 out of m models with the worst target accuracy is chosen.  This interpretation, however, makes the high numbers in table 1, like 80/131, very surprising given the value of m.

**Documentation:**

N/A

**Limitations:**

The authors have brought up the high cost of fully training all candidate models, but handwaved it as part of the model selection setting.

**Opportunities For Improvement:**

While the proposed method is effective, it is too straightforward and simple.  Maybe you can iteratively refine the candidate pool by removing the worst performing model to try and select the best-case model.

Additionally, despite the claim that fully training a bunch of candidate models is baked into the model selection setting, I still think you should have analysis of how the method performs with a limited number of candidates. How well does the ensemble approach work with only 3, 5, 7 etc. numbers of candidates compared other methods?

**Relation To Prior Work:**

Yes

**Summary And Contributions:**

This paper proposes an ensemble based method for model selection in UDA tasks that can reliably avoid choosing the worst model and consistently outperforms competing model selection methods on various UDA benchmarks.

---

> ### Author Rebuttal · Authors · 2024-08-16
>
> We appreciate your constructive feedback. We have provided comprehensive responses to all of your questions. We are available for further discussion to address any additional concerns you may have.
>
> Firstly, to avoid any misunderstandings, we **respectfully emphasize that our submission is for the Datasets and Benchmarks Track rather than the main track**. As indicated in the title of our submission, we have **two main contributions**: 1) we conduct **a comprehensive empirical study** and uncover **a surprising phenomenon** where existing model selection approaches fail to avoid worst-case scenarios effectively, and 2) we propose **a simple yet certified baseline method** to reliably address this problem, serving as a versatile baseline for various domain adaptation settings. With both contributions, we **aim to inspire future efforts in developing more reliable model selection methods**.
>
> We are pleased to see that **both Reviewer uZP5 and Reviewer yaswW positively acknowledged the significance of both the benchmarking and the baseline method**.
>
> We have noted that your review **seems to focus solely on the proposed baseline method, potentially overlooking the benchmarking aspect**. Before addressing your point-by-point feedback, we wanted to clarify this matter.
>
> > **Q1**: **While the proposed method is effective, it is too straightforward and simple.** Maybe you can iteratively refine the candidate pool by removing the worst performing model to try and select the best-case model.
>
> **A1**: We **respectfully disagree** with the view that the proposed method is too straightforward and simple. We believe that if a simple and straightforward method is effective, then **additional complexity may not be necessary**. In AI research, many well-known methods are indeed very simple and straightforward, such as residual connections, dropout, and attention mechanisms. Our EnsV method is designed to serve as **a versatile baseline** for various domain adaptation settings, and **its simplicity is advantageous for comparison and innovation by future researchers.**
>
> We **appreciate your constructive suggestion to iteratively refine** our EnsV by removing the worst-performing model. In fact, we have previously considered this strategy. However, we did not adopt it for two main reasons: 1) To ensure that the iterative approach improves performance, **the number of iterations** would need to be determined as **a new hyperparameter**; and 2) The iterative method would introduce **additional complexity** and efficiency concerns, as noted by the **Reviewer uZP5**.
>
> We selected tasks where there is a performance gap between our EnsV and the best results. We experimented with different numbers of iterations (0, 1, 2, 3). In each iteration, we removed the worst-scoring candidate model according to our role model (the Ensemble) and prepared for the next iteration. Our default EnsV does not include any iterations (i.e., iteration count is 0). **We included the results in Table 1 of the PDF.** While iterations may offer some benefits, they introduce the **Iteration Count as a new hyperparameter**, which we prefer to avoid. Therefore, **as a baseline method for hyperparameter selection, we adopted the simpler design without iterations and thus without additional hyperparameters.**
>
> > **Q2**:
> The authors have brought up the high cost of fully training all candidate models, but handwaved it as part of the model selection setting. Additionally, despite the claim that fully training a bunch of candidate models is baked into the model selection setting, I still think you should have analysis of how the method performs with a limited number of candidates. How well does the ensemble approach work with only 3, 5, 7 etc. numbers of candidates compared other methods?
>
> **A2**: The inclusion of all candidate models in the model selection problem setting is **standard in existing domain adaptation research and not something we specifically curated or claimed. We adhere to this standard to ensure fair benchmarking**.
>
> In our experiments, we have **evaluated 7 models for single-hyperparameter tasks** and up to 28 models for classification and 48 models for segmentation tasks in the case of two-hyperparameter tasks. To further **investigate the impact of the number of candidate models**, we considered a challenging scenario where **the pool initially contains only the best candidate model and gradually added an increasing number (k) of inferior models based on ascending target-domain accuracy**. We experiment with the task MDD on Ar$\to$Cl, adding from k==1 to k==27 (all inferior models) to the poor. Results from **Figure 1 in the PDF** demonstrate that **EnsV consistently avoids selecting the worst or extremely inferior models**, whether with a small number of candidates (e.g., 3, 5) or a larger number of suboptimal models (e.g., 15, 20).
>
> > **Q3**: The paper could use a little more clarification on the worst-case analysis in table 1. I am assuming the worst case is when the 1 out of m models with the worst target accuracy is chosen. This interpretation, however, makes the high numbers in table 1, like 80/131, very surprising given the value of m.
>
> **A3**: Your understanding is correct, and we appreciate the feedback. We will add more clarifications in our revision. The high numbers in Table 1, such as 80/131, **highlight a surprising but overlooked issue**: despite high peak performance claims, existing model selection methods often fail to consistently avoid the worst model and sometimes select it frequently. This insight emphasizes that for reliable model selection in real-world applications, **it's crucial to avoid the worst-case scenarios, not just focus on peak or average performance**. We provide **specific examples** in Table 1 of our appendix to illustrate cases where **existing methods select the worst model.** For reference, **we include the results in Table 2 of the PDF**.

---

> ### Comment · Reviewer_deqf · 2024-09-01
> **Response to Rebuttal**
>
> Looking at your rebuttal and the other reviews, I agree that I overlooked the benchmarking aspect of this paper when writing my review.
>
> Regarding the baseline method, I still have concerns. Because model ensembling is a well-established technique known for reducing prediction errors across various tasks, its contribution as a baseline is limited. It is not just the simplicity of the method, but also the novelty is extremely lacking.
>
> I am also not convinced by the model selection problem setting.  I understand needing to follow the standard problem setting when comparing to prior work, but this can be done in conjunction with addressing the problems inherent in the the standard model selection setting.  For example, because closed-set UDA did not reflect the challenges of domain adaptation in practice, people proposed source-free DA, open-set DA, universal DA, test time adaptation, etc.  Right now, the proposed EnsV method relies on the so called "free lunch" of having access to a collection of fully trained candidate models, but this setting is impractical for actual application in hyperparameter tuning for domain adaptation due to the huge computation cost.
>
>
> When considering the benchmarking aspect and the new experimental results in the rebuttal, I change my recommendation from  4: borderline reject to 5: borderline accept.
>
> *correction, from 4 to 6: Marginally above acceptance threshold

---

> > ### Author Response · Authors · 2024-09-01
> > **Thanks for raising the score to boarderline acceptance.**
> >
> > We appreciate the reviewer’s updated feedback and the change in **recommendation to borderline accept**.
> >
> > We are pleased that **our clarification has helped the reviewer recognize our contribution** to the empirical study of the model selection problem, which **was initially overlooked in the review**.
> >
> > Regarding the baseline method, we **addressed concerns about its simplicity** in our initial rebuttal. We maintain that **simplicity is a strength but never a weakness in a baseline method**, as it allows for universal and straightforward implementation across various model selection problems.
> >
> > We note that **the reviewer has raised a new concern about novelty, which was not mentioned in the initial review**. We are happy to clarify the novelty of our submission. While ensemble methods themselves are not new, our approach introduces **a novel application of ensemble techniques** within the context of model selection, supported by a theoretical analysis of worst-case scenarios. Additionally, our work provides **a significant novel contribution to benchmarking**. We identify and address a previously **overlooked shortcoming** in current model selection methods: avoiding the selection of worst-case or extremely poor-performing models.
> >
> > Last but not least, regarding the concern about the model selection problem setting, we are pleased that our rebuttal has helped **the reviewer recognize that this setting is a standard one widely studied by the model selection community and should not be viewed as a specific weakness of our submission**. We agree that **exploring new problem settings could be valuable**, especially if they could be more practical. However, we emphasize that this **should not diminish the significance and necessity of our work**. The current model selection community needs a comprehensive and fair benchmarking study to reveal both progress and unresolved issues. Our paper offers **a novel perspective by investigating the worst-case selection** within existing model selection methods and provides a simple yet effective baseline method. Furthermore, this baseline is applicable across various model selection contexts, including source-free and open-set domain adaptation, which have been **rarely or never explored in existing model selection research**.
> >
> > In conclusion, we sincerely thank the reviewer for their further reply and for raising the score to borderline accept. We are always open to further discussions.

---

### Official Review · Reviewer_uZP5 · 2024-07-25

**Rating:** 7
**Confidence:** 2

**Review:**

Quality: The paper is of high quality, providing a thorough empirical evaluation of multiple model selection methods and proposing a robust new method, EnsV. The methodology is rigorous, and the experiments are comprehensive.

Clarity: The paper is well-written and clear, though some sections with technical details could benefit from additional explanation for broader accessibility.

Originality: The work is original, addressing the under-explored area of reliable model selection in UDA and introducing a novel ensemble-based approach that avoids worst-case selections.

Significance: The benchmark and the proposed EnsV method are significant contributions, offering practical solutions to a critical issue in UDA. This can potentially impact the deployment of UDA methods in real-world applications.

**Strengths:**

The paper conducts a thorough empirical study across multiple UDA methods and benchmarks.

Introduction of the EnsV method, which effectively avoids worst-case model selections.

EnsV is supported by both theoretical analysis and extensive empirical results.

The proposed method addresses practical challenges in UDA model selection, making it highly relevant for real-world applications.

**Additional Feedback:**

N/A

**Clarity:**

The paper is well-written, clearly presenting the methodology, results, and implications. The structure is logical, and the figures and tables effectively support the text.

**Correctness:**

The claims made in the submission are correct, supported by thorough empirical and theoretical validation. The evaluation methods and experimental design are appropriate and performed correctly.

**Documentation:**

The paper includes sufficient detail to support reproducibility. The experimental setup, datasets, and evaluation protocols are well-documented, allowing other researchers to replicate the study.

**Limitations:**

The authors have addressed the limitations of existing model selection methods.

**Opportunities For Improvement:**

Reducing the computational requirements of maintaining and evaluating multiple models could make EnsV more practical for resource-constrained environments.

Addressing scalability concerns by optimizing EnsV for large datasets and numerous candidate models would enhance its applicability in large-scale UDA tasks.

**Relation To Prior Work:**

The paper clearly discusses how it builds upon and differs from previous contributions. It positions the proposed EnsV method within the context of existing model selection approaches in UDA, highlighting its unique advantages.

**Summary And Contributions:**

The paper investigates the reliability of existing model selection methods for Unsupervised Domain Adaptation (UDA) and introduces a novel ensemble-based selection approach named EnsV. Through a comprehensive empirical study involving 8 existing model selection approaches evaluated across 12 UDA methods and 5 diverse benchmarks, the authors find that none of these approaches consistently avoid worst-case selections. EnsV, however, is shown to empirically and theoretically avoid the worst-case selection, proving to be versatile and reliable across various UDA scenarios. This study emphasizes the importance of considering worst-case avoidance in model selection and calls for more attention to the reliability of model selection methods in UDA.

---

> ### Author Rebuttal · Authors · 2024-08-16
>
> We sincerely appreciate your comprehensive and positive comments on our submission, especially for **acknowledging our significant contributions to both the benchmarking and the simple yet versatile baseline method**.
>
> > **Q1**: Reducing the computational requirements of maintaining and evaluating multiple models could make EnsV more practical for resource-constrained environments.
>
> **A1**: **We agree** that addressing the computational challenges of maintaining and evaluating multiple models is crucial, especially in resource-constrained environments. To our knowledge, no existing work specifically focuses on this aspect. We recognize that maintaining and applying multiple models for inference in such scenarios can be costly, and we hope that future research will give more attention to this practical problem.
>
> For our submission, we want to clarify that **we adhere to the standard model selection setting that does not specifically consider resource efficiency**. In this setting, people can assume that **multiple models have already been trained**, and the goal is to select the best one from these pre-trained models. Notably, compared to existing model selection methods, our proposed baseline EnsV is already simpler and more practical: it does not require access to source information, does not necessitate training new models, does not introduce new hyperparameters, and only requires the inferred predictions from all candidate models to make a reliable selection. This makes EnsV versatile and applicable to various domain adaptation settings.
>
>
> > **Q2**: Addressing scalability concerns by optimizing EnsV for large datasets and numerous candidate models would enhance its applicability in large-scale UDA tasks.
>
> **A2**: **We agree** that enhancing scalability is crucial for making EnsV more practical for real-world applications.
>
> For **large datasets**, we included DomainNet, one of the largest domain adaptation benchmarks as acknowledged by Reviewer N1tk, in our experiments. However, we recognize that DomainNet may not fully represent the scale of real-world applications.
>
> Regarding the handling of **numerous candidate models**, our theoretical analysis suggests that EnsV can effectively avoid worst-case scenarios. In our two-hyperparameter experiments, discussed in Lines 263-266, we evaluated EnsV with 28 models for classification tasks and 48 models for segmentation tasks, demonstrating competitive performance.
>
> We appreciate this comment and want to clarify that while we have considered scalability within the existing model selection framework, **we acknowledge that there is much more to be done. We look forward to future advancements** that will optimize model selection for even larger datasets and more candidate models.

---

### Official Review · Reviewer_N1tk · 2024-07-25
**Review for Towards Reliable Model Selection for Unsupervised Domain Adaptation: An Empirical Study and A Certified Baseline**

**Rating:** 5
**Confidence:** 4
**Correctness:** Yes
**Clarity:** Yes

**Review:**

Pros:
- This paper provides a benchmark for DA methods analysis
- This paper delivers a parameter validation solution
- This paper gives the idea of choosing DA methods.

Cons:
- The datasets used in the benchmark are not representative. For example, in computer vision, the dataset sizes of Office31 and OfficeHome are very small and cannot adequately represent the field. The authors should consider using DomainNet for a more comprehensive analysis. Results from Office31 and OfficeHome are insufficient to fully represent the domain adaptation (DA) problem in computer vision.
- Deep learning has been widely applied across various research areas, and analyzing "shallow" DA methods within a deep learning framework may render the conclusions impractical. For computer vision datasets, the authors should analyze the impact of different vision encoders, such as ResNet, Vision Transformers, and VGGNet.
- I would like to see if the conclusions drawn from "shallow" DA methods would still hold when applied to "deep" DA methods.

**Strengths:**

see pros in section "review"

**Additional Feedback:**

see the section review.

**Documentation:**

Yes

**Limitations:**

Yes

**Opportunities For Improvement:**

see cons in section "review"

**Relation To Prior Work:**

Yes

**Summary And Contributions:**

This paper proposes a benchmark to evaluate existing shallow domain adaptation methods from various shift aspects. The benchmark includes datasets from computer vision, natural language processing, tabular data, and biomedical domains. It introduces a nested loop cross-validation procedure for parameter validation and provides guidelines for selecting appropriate scorers. The experimental results offer valuable insights for choosing domain adaptation models.

---

> ### Author Rebuttal · Authors · 2024-08-16
>
> We appreciate the time and effort the reviewer has devoted to evaluating our paper. However, after a thorough and detailed comparison between the review and our submission, **we respectfully believe that this review may be intended for a different NeurIPS submission.**
>
> Several pieces of evidence:
>
> - The comments in the **Summary and Contributions** section do not align with our paper. Specifically, **we did not use** “datasets from natural language processing, tabular data, and biomedical domains”, **nor did we propose** “a nested loop cross-validation procedure for parameter validation”.
>
> - The criticisms in the **Cons section** contain factual inaccuracies. Our submission **already addresses the points raised** by the reviewer. Detailed **point-by-point responses** are provided below.
>
> > **Q1**: The datasets used in the benchmark are not representative. For example, in computer vision, the dataset sizes of Office31 and OfficeHome are very small and cannot adequately represent the field. The authors **should consider using DomainNet for a more comprehensive analysis.** Results from Office31 and OfficeHome are insufficient to fully represent the domain adaptation (DA) problem in computer vision.
>
> **A1**: We agree that using large-scale benchmarks such as DomainNet is important for domain adaptation research. In our original submission, we **already included results on DomainNet (see Table 6)** and observed that our proposed baseline, EnsV, demonstrates a greater performance advantage on DomainNet.
>
> > **Q2**: Deep learning has been widely applied across various research areas, and **analyzing "shallow" DA methods** within a deep learning framework may render the conclusions **impractical**. For computer vision datasets, the authors **should analyze the impact of different vision encoders, such as ResNet, Vision Transformers, and VGGNet.**
>
> **A2**: We agree that analyzing the impact of different deep learning backbones is important. In our study, we **already evaluated** widely used backbones in domain adaptation, including ResNet and **Vision Transformers**. The impact of these backbones is discussed in **Lines 267-271** of the manuscript. Notably, the results in **Table 11** demonstrate that EnsV is significantly more robust with Vision Transformers compared to other model selection methods.
>
> > **Q3**: I would like to see if **the conclusions drawn from "shallow"** DA methods would still hold when **applied to "deep" DA methods.**
>
> **A3**: Till this question, we suspect there may be a misunderstanding or, more likely, that this review was intended for a different NeurIPS submission. The evidence for this is that all of our experiments and analyses are based on deep domain adaptation (DA) methods, and **we have not mentioned or used "shallow" DA methods** in our paper.

---

### Author Rebuttal · Authors · 2024-08-16

We sincerely appreciate the reviewers for their valuable feedback. We respectfully point out that **the review by Reviewer N1tk may pertain to a different submission** focused on shallow domain adaptation methods rather than our paper. Specifically, we are grateful for the **following acknowledgments of our submission**:

- **Our studied problem** of model selection in domain adaptation is: under-explored but highly relevant for real-world applications [Reviewer uZP5], important [Reviewer deqf], and significant [Reviewer yasW].

- **Our empirical study** and observations are described as thorough and comprehensive, significant and impactful [Reviewer uZP5], very surprising [Reviewer deqf], and extensive, fundamental, and significant [Reviewer yasW].

- **Our proposed baseline method**, EnsV, is noted as: supported by both theoretical analysis and extensive empirical results [Reviewer uZP5], effective yet straightforward and simple [Reviewer deqf], and fundamental, simple, and easy to implement [Reviewer yasW].

These acknowledgments encourage us to continue exploring this under-explored problem in the future.

---

### Decision · Program_Chairs · 2024-09-26

**Decision:**

Accept (Poster)

**Comment:**

This paper studies the model selection in unsupervised domain adaption. The problem is overlooked by the previous study. Therefore, the authors conduct a comprehensive benchmarking of 8 model selection strategies for 12 UDA methods across 5 diverse UDA benchmarks and 5 UDA scenarios. They find that none of the benchmarked approaches is able to avoid the worst-case selection, and propose a simple ensemble based strategy to mitigate the issue.

Specifically, the model selection is a crucial yet overlooking issue in successful UDA. The importance of this issue and the contribution in revealing the issue through comprehensive benchmarking is recognized by most reviewers. However, some concerns regarding the simplicity and the computational overhead of the proposed strategy are raised. The discussion of the limitations and the additional experiments are suggested to include during revision. Meanwhile, it is also suggested to include the discussion of the model selection in general domain generalization to facilitate readers better understand the position of this work:
- Ensemble of Averages: Improving Model Selection and Boosting Performance in Domain Generalization, NeurIPS’22.
- Pareto Invariant Risk Minimization, ICLR’23.

Upon checking the review comments and the discussion between reviewers and the authors, the strengths in benchmarking overweighs the methodological limitation. As a submission to the benchmark track of NeurIPS, it is above the bar and should be accepted.